# Reactive Transport with Wellbore Storages in a Single-Well Push-Pull Test

Quanrong Wang[1,2,3] and Hongbin Zhan[3*]

[1]Laboratory of Basin Hydrology and Wetland Eco-restoration, China University of Geosciences, Wuhan, Hubei, 430074, P. R. China

[2]School of Environmental Studies, China University of Geosciences, Wuhan, Hubei, 430074, P. R. China

[3]Department of Geology and Geophysics, Texas A& M University, College Station, TX 77843-3115, USA

*Correspondence to*: Hongbin Zhan (zhan@geos.tamu.edu)

**Abstract:** Using the single-well push-pull (SWPP) test to determine the in situ biogeochemical reaction kinetics, a chase phase and a rest phase were recommended to increase the duration of reaction, besides the injection and extraction phases. In this study, we presented multi-species reactive models of the four-phase SWPP test considering the wellbore storages for both groundwater flow and solute transport and a finite aquifer hydraulic diffusivity, which were ignored in previous studies. The models of the wellbore storage for solute transport were proposed based on the mass balance, and the sensitivity analysis and uniqueness analysis were employed to investigate the assumptions used in previous studies on the parameter estimation. The results showed that ignoring it might produce great errors in the SWPP test. In the injection and chase phases, the influence of the wellbore storage increased with the decreasing aquifer hydraulic diffusivity. The peak values of the breakthrough curves (BTCs) increased with the increasing aquifer hydraulic diffusivity in the extraction phase, and the arrival time of the peak value became shorter with a greater aquifer hydraulic diffusivity. Meanwhile, the Robin condition performed well at the rest phase only when the chase concentration was zero and the solute in the injection phase was completely flushed out of the borehole into the aquifer. The Danckwerts condition was better than the Robin condition even when the chase concentration was not zero. The reaction parameters could be determined by directly best fitting the observed data when the non-linear reactions were described by piece-wise linear functions, while such an approach might not work if one attempted to use non-linear functions to describe such non-linear reactions. The field application demonstrated that the new model of this study performed well in interpreting BTCs of a SWPP test.

## 1. Introduction

Single-well push-pull (SWPP) test is a popular technique to characterize the in situ geological formations and to remedy the polluted aquifer by a series of biogeochemical reactions (Istok, 2012; Phanikumar and McGuire, 2010; Schroth and Istok, 2006). Therefore, the accuracy of the results is not only dependent on the experimental operation, but also on the conceptual model which is expected to properly represent the physical and biogeochemical processes. Unfortunately, most previous studies of the multi-species reactive transport models were based on some assumptions which may not be satisfied in actual applications, although those assumptions usually simplified the mathematical treatment of the problem (Istok, 2012; Wang et al., 2017).

As for the analytical solutions of the SWPP test, they have been widely used for applications, due to the high efficiency and great accuracy of the solutions, like the model of Gelhar and Collins (1971) for a fully penetrating well, the model of Schroth and Istok (2005) for a point source/sink well, and the model of Huang et al. (2010) for a partially penetrating well, assuming that the advection, the dispersion and the first-order reaction were involved in the transport processes. Haggerty et al. (1998) and Snodgrass and Kitanidis (1998) presented a simplified method based on a well-mixed reactor to estimate the first-order and zero-order reaction rate, without involving complex numerical modelling. Schroth and Istok (2006) provided two alternative models, one of them was a plug-flow model and the other was a variably mixed reactor model. Schroth et al. (2000) presented a simplified method for estimating retardation factors, based on the model of Gelhar and Collins (1971). Istok et al. (2001) extended the models of Haggerty et al. (1998) and Snodgrass and Kitanidis (1998) to estimate the Michaelis-Menten kinetic parameters which were used to describe the microbial respiration in the aquifer. Jung and Pruess (2012) presented a closed-form analytical solution for heat transport in a fractured aquifer involving a push-and-pull procedure. However, the mentioned-above analytical or semi-analytical solutions of the SWPP test were based on some over-simplified assumptions. For instance, the hydraulic diffusivity of the aquifer was assumed to be infinite, resulting in a time-independent flow velocity, where the hydraulic diffusivity is the ratio of the radial hydraulic conductivity over the specific storage. The wellbore storage effect on the flow field was assumed to be negligible as well. Therefore, how accurate could parameter estimation be needs to be tested? Recently, Wang et al. (2017) investigated the influences of a finite hydraulic diffusivity on the results and found that it might be significant, since both advective and dispersive transport were related to the flow velocity. One point to note is that the model of Wang et al. (2017) still contains an additional issue that has not been addressed: the wellbore storage influence on solute transport, which will be the focal point of this investigation.

The wellbore storage for solute transport refers to the variation of the solute injected in the wellbore during the processes of the test. A complete SWPP test contains four principle phases: injection of a prepared solution (tracer) into a targeted aquifer; injection of a chaser; rest period; extraction of the mixture solution. The second and third phases are optional but are recommended to extend the reaction time of the tracer in the aquifer. In the injection phase, the concentration of the solute in the wellbore is smaller than that of the original solution at the early stage, since the original solute could be diluted by the original water in the wellbore, due to the mixing effect. Therefore, excluding the wellbore

storage may overestimate the concentration in the wellbore at the early stage of the injection phase before the pre-test water inside the wellbore was completely flushed out of the borehole into the aquifer. In the chaser phase, the concentration of the solute in the wellbore may be greater than the concentration of the chaser, due to the mixing effect. The treatment of excluding the wellbore storage could underestimate the concentration in the wellbore at the early stage of the chase phase, due to the high concentration of solute in the wellbore at the end of the injection phase. When the chaser phase is absent or the chaser concentration is not zero, the concentration might not be zero in the early stage of the rest phase. As for the chaser concentration, it is usually set as zero. However, under some circumstances, investigators may use a non-zero concentration for the chase phase. For example, Phanikumar and McGuire (2010) used 10 mg/L for $Cl^-$ and 2 mg/L for $SO_4^{2-}$ in their chase solutions. Therefore, the concentration at the well screen may not be zero at the early stage of the rest phase when the chase concentration was not zero. All these mixing effects occurring in the wellbore is named wellbore storage of the solute transport. Obviously, the assumption of ignoring the wellbore storage is not reasonable for the solute transport.

Actually, the above-mentioned assumptions used in the analytical and semi-analytical solutions can be relaxed in the numerical models, such as MODFLOW/MT3DMS (Harbaugh et al., 2000; Zheng and Wang, 1999), FEFLOW (Diersch, 2014), SUTRA (Voss, 1984), STOMP (Nichols et al., 1997), and so on. Huang et al. (2010), Sun (2016), Haggerty et al. (1998), and Schroth and Istok (2006) respectively employed such four software packages to carry out numerical simulations of SWPP tests, mainly involving advection, dispersion and first-order reaction. Unfortunately, the traditional three-dimensional models in the Cartesian coordinate system may create some errors in describing the wellbore storage of solute transport in the wellbore-confined aquifer, which is explained in Supplemental Materials.

This study addresses multi-species reactive transport associated with SWPP tests with a better conceptual model that acknowledges the realistic circumstances that have been either overlooked or overly simplified in previous investigations. Firstly, we will employ a more realistic finite hydraulic diffusivity instead of an infinite hydraulic diffusivity to describe the flow field. Secondly, we will propose a better way to handle the boundary condition of transport at the wellbore by consider the wellbore storage effect for both groundwater and solute transport during the SWPP tests. Thirdly, the new model is tested using a field test dataset reported in McGuire et al. (2002). Fourthly, the sensitivity analysis and uniqueness analysis will be employed to investigate the assumptions used in previous studies on the parameter estimation.

## 2. Problem statement of the SWPP test

A cylindrical coordinate system is adopted with the $r$-axis horizontal and the $z$-axis vertically upward, as shown in Figure 1. The origin is at the centre of the well and located in the plane of symmetry of the aquifer. The well fully penetrates a confined aquifer with a constant thickness. The aquifer is homogeneous, and the influence of the regional flow could beis ignored.

## 2.1 Revisit of the previous model

The general form of the governing equation for a multi-species reactive SWPP test is

$$\frac{\partial C_i}{\partial t} + \frac{\rho_b}{\theta}\frac{\partial S_i}{\partial t} = -\sum_{j=1}^{N-1}[\mathcal{H}(t - t_j^*) - \mathcal{H}(t - t_{j+1}^*)]\lambda_j C_i^{n_j} \pm F_j, t > 0, \tag{1}$$

where $C_i$ is the aqueous phase concentration of the $i^{th}$ reactive solute, $S_i$ is the solid phase concentration of the $i^{th}$ reactive solute, $t$ is the time in the SWPP test, $\rho_b$ is the bulk density, $\theta$ is the porosity, $\mathcal{H}$ is the Heaviside step function, $\lambda_j$ and $n_j$ are the constant and orders, $N$ is the number of the segment, $t_j^*$ and $t_{j+1}^*$ are the times at two ends of segment $j$, and $F_j$ Monod/Michaelis-Menten kinetics. For the purpose of the simplicity, we only present the reactive processes of the chemicals as described by Eq. (1), while the expressions of the transport (e.g. dispersion, diffusion, and advection) could be seen in Phanikumar and McGuire (2010) which used it to describe biogeochemical reactive transport of an arbitrary number of species including Monod/Michaelis-Menten kinetics, and the sorption models could be isotherm (Freundlich, Langmuir and linear sorption), one-site kinetic and two-site kinetic. As for the studies of Gelhar and Collins (1971), Schroth and Istok (2005), Huang et al. (2010), Haggerty et al. (1998), Snodgrass and Kitanidis (1998), Schroth and Istok (2006), Schroth et al. (2000), Istok et al. (2001), Jung and Pruess (2012), Wang et al. (2017), and so on, the governing equation is a special case of Eq.(1).

As mentioned in Introduction, several assumptions may be debatable in previous studies and could be the source of errors for the actual applications. Firstly, the transport model is composed of a set of advection-dispersion equations (ADEs) built on the basis of flow velocity which is assumed to be time-independent (Chen et al., 2017; Gelhar and Collins, 1971; Huang et al., 2010; Phanikumar and McGuire, 2010):

$$v_r = \frac{Q}{2\pi rB\theta}, r \geq r_w, \tag{2}$$

where $r_w$ is the well radius; $r$ is the radius distance from the centre of the well; $B$ is the aquifer thickness; $Q$ is the flow rate of the well; $v_r = u_r/\theta$ is the average radial pore velocity and $u_r$ is the radial Darcian velocities. Eq. (2) implies that the hydraulic diffusivity of the aquifer is infinite, thus the flow velocity is independent of time. Meanwhile, the wellbore storage is negligible or the well radius $r_w$ is assumed to be infinitesimal in formulating Eq. (2).

The second assumption of the model is the boundary condition of the well screen in the rest phase of the SWPP test, in which a Robin condition (or a third-type condition) is employed to describe the aqueous solute transport (Chen et al., 2017; Phanikumar and McGuire, 2010; Wang et al., 2017):

$$\left(v_r C - D_r \frac{\partial C}{\partial r}\right)\Big|_{r \to r_w} = 0, t_{inj} + t_{cha} < t \leq t_{inj} + t_{cha} + t_{res}, \tag{3}$$

where $t_{inj}, t_{cha}, t_{res}$, and $t_{ext}$, represent the durations of the injection, chase, rest, and extraction phases, respectively; $C$ is resident concentration of the aqueous phase to represent $C_i$ in Eq. (1); $D_r$ is the dispersion coefficient, which is

$$D_r = \alpha_r v_r + D_0, \tag{4}$$

in which $\alpha_r$ is the radial dispersivity; $D_0$ is the effective diffusion coefficient in the aquifer.

Thirdly, a constant solute concentration in the wellbore is applied in the injection and chase phases without considering the solute diluted effect in the wellbore (Chen et al., 2017; Gelhar and Collins, 1971; Istok, 2012; Phanikumar and McGuire, 2010; Wang et al., 2017)

$$\left(v_r C - D_r \frac{\partial C}{\partial r}\right)\bigg|_{r \to r_w} = v_r C_0^{inj}, \, 0 < t \le t_{inj}, \tag{5a}$$

$$\text{or } C|_{r \to r_w} = C_0^{inj}, \, 0 < t \le t_{inj}, \tag{5b}$$

$$\left(v_r C - D_r \frac{\partial C}{\partial r}\right)\bigg|_{r \to r_w} = v_r C_0^{cha}, \, t_{inj} < t \le t_{inj} + t_{cha}, \tag{6a}$$

$$\text{or } C|_{r \to r_w} = C_0^{cha}, \, t_{inj} < t \le t_{inj} + t_{cha}, \tag{6b}$$

where $C_0^{inj}$ and $C_0^{cha}$ represent the solute concentrations injected into the wellbore during the injection and chase phases, respectively. A detailed discussion about mentioned-above assumptions can be seen in Phanikumar and McGuire (2010) or Wang et al. (2017).

Fourthly, the solute transport caused by dispersion and advection was assumed to be negligible in estimating the reaction rates. For instance, one of the simplest models of such reactions may be the first-order reaction

$$\frac{\partial C}{\partial t} = -\lambda C, \tag{7}$$

where $\lambda$ is the first-order reaction rate constant. Beside the first-order reaction, Eq. (7) could be used to describe the first-order biodegradation and radioactive decay. Haggerty et al. (1998) presented a simplified method to estimate $\lambda$ for the SWPP test:

$$\ln\left(\frac{C_{rec}(t^*)}{C_{tra}(t^*)}\right) = \ln\left(\frac{1 - exp(-\lambda t_{inj})}{\lambda t_{inj}}\right), \tag{8}$$

where $t^*$ is time since the end of injection; $C_{rec}(t^*)$ is the reactant concentration; $C_{tra}(t^*)$ is the concentration of a conservative tracer. To obtain the value of $\lambda$, the reactant and the conservative tracer should be fully mixed and injected into the aquifer simultaneously to conduct the SWPP test. After measuring the data of $C_{rec}(t^*)$ and $C_{tra}(t^*)$ in the extraction phase, one could fit the data of $\ln\left(\frac{C_{rec}(t^*)}{C_{tra}(t^*)}\right) \sim t^*$ using a linear function and the slope of $t^*$ is the estimation of $\lambda$. Snodgrass and Kitanidis (1998) derived a similar model for estimating $\lambda$:

$$\ln\left(\frac{C_{rea}(t^*)}{C_{tra}(t^*)}\right) = \ln\left(\frac{C_{rec}^0}{C_{tra}^0}\right) - \lambda t^*. \tag{9}$$

Comparing Eq. (8) with Eq. (9), one could find that the difference is the first terms on the right sides of equations, while $\lambda$ is the slope for both Eqs. (8) and (9). Although the accuracy of both models has been tested by a number of investigators, previous studies on reactive transport were based on an assumption that the aquifer hydraulic diffusivity was infinite (e.g. Eq. (1) of Reinhard et al. (1997) , and Eq. (2) of Haggerty et al. (1998)).

Actually, the assumptions of Eqs. (2) - (9) are debatable for the actual applications, and may cause errors in modelling the solute transport in the SWPP test. The second and third assumptions relate to the wellbore storage of the solute transport in the SWPP test. In the following section, the new models will be proposed to investigate the potential errors when these assumptions are involved.

## 2.2 A revised model with a finite hydraulic diffusivity

As for the first assumption in Section 2.1, Wang et al. (2017) demonstrated that it might result in non-negligible errors in parameter estimation, particularly for the estimation of dispersivity. A minor point to note is that the model of Wang et al. (2017) mainly focused on conservative solute transport, rather than reactive transport. Nevertheless, the pore velocity of transient flow is calculated by Darcy's law

$$v_r = \frac{K_r}{\theta} \frac{\partial s}{\partial r}, \tag{10}$$

where $K_r$ is the radial hydraulic conductivity; $s$ is drawdown which could be obtained by solving the following mass balance equation with the proper initial and boundary conditions

$$\frac{\partial v_r}{\partial r} + \frac{v_r}{r} = \frac{S_s}{\theta} \frac{\partial s(r,t)}{\partial r}, \ r \geq r_w, \tag{11}$$

$$s(r,t)|_{t=0} = 0, \tag{12}$$

$$v_r|_{r \to \infty} = 0, \tag{13}$$

$$(2\pi B v_r)|_{r \to r_w} - \frac{\pi r_w^2}{\theta} \frac{ds_w(t)}{dt} = Q, \tag{14}$$

where $S_s$ is the specific storage of aquifer; $s_w$ is the drawdown inside the wellbore.

        As for the second assumption in the rest phase, as shown Eq. (3), it implies that the concentration of solute is zero in the wellbore. This assumption works when the chase concentration is zero and the prepared solution is completely pushed out of the borehole into the aquifer at the end of the chase phase. However, the chase concentration might be not zero, as

demonstrated in Phanikumar and McGuire (2010) and McGuire et al. (2002). Consequently, the concentration in the early stage of the rest phase, which is close to the concentration at the end of the chase phase, is not zero. This is because the water level in the wellbore is greater than the hydraulic head in the surrounding aquifer due to the wellbore storage, resulting in a positive flux from the wellbore into the aquifer. Correspondingly, when the chase concentration is not zero or the prepared

solution in the injection phase is not completely pushed out of the wellbore, the concentration in the wellbore may not be zero in the early stage of the rest phase. In this study, we employed the Danckwerts condition for transport at the well screen in the rest period (Danckwerts, 1953)

$$\frac{\partial C}{\partial r}\bigg|_{r \to r_w} = 0, \ t_{inj} + t_{cha} < t \le t_{inj} + t_{cha} + t_{res}. \tag{15}$$

Actually, Eq. (15) acknowledges the continuity of concentration and continuity of mass flux simultaneously across the well screen, namely, $C|_{r \to r_w^-} = C|_{r \to r_w^+}$ and $(v_r C)|_{r \to r_w^-} = \left(v_r C - D_r \frac{\partial C}{\partial r}\right)\bigg|_{r \to r_w^+}$, where $-$ and $+$ signs in the subscript of $r_w$ represent approaching the well screen from inside the well and outside the well, respectively.

The third assumption mentioned in Section 2.1 seems not reasonable at the early stage of the injection and chase phases, because the concentration of the injected solute will be affected by the finite volume of water in the wellbore. Take the chase

phase as an example: it is impossible to immediately reduce the solute concentration inside the wellbore from a certain level during the tracer injection phase to zero when switching to the chase phase, even when the solute concentration in the chase phase is zero. This is because the wellbore with a finite radius contains a certain finite mass of solute at the moment of switching from injection of a tracer to injection of a chaser. Therefore, it will take some time to completely flush out the residual tracer inside the wellbore after the start of the chase phase, and a larger wellbore will take a longer time to flush out

the residual tracer inside the wellbore. This means that the concentration at the wellbore/aquifer interface will not drop to zero immediately after the start of the chase phase. Instead, it will take a finite period of time to gradually approach zero during the chase phase. Similarly, the boundary condition of the well screen in the injection phase might not be appropriate in previous studies if the wellbore storage effect is of a concern. Therefore, the value of solute concentration inside the wellbore should be smaller or equal to $C_0^{inj}$ in the injection phase and greater or equal to $C_0^{cha}$ in the chase phase.

Here, we will develop a new approach to take care of the concentration in the wellbore in the injection and chase phases based on the mass balance principle, i.e.,

$$\Delta m = C_0^{inj} Q \Delta t = C_w^{t+\Delta t}(V^t + Q\Delta t) - C_w^t V^t, \ 0 < t \le t_{inj}, \tag{16}$$

$$\Delta m = C_0^{cha} Q \Delta t = C_w^{t+\Delta t}(V^t + Q\Delta t) - C_w^t V^t, \ t_{inj} < t \le t_{inj} + t_{cha}, \tag{17}$$

where $\Delta m$ represents the mass entering into the well during time interval $\Delta t$; $C_w^t$ and $C_w^{t+\Delta t}$ represent the solute

concentrations in the wellbore at the time $t$ and $t + \Delta t$, respectively; $V^t$ represents the volume of water in the wellbore at the time $t$. The initial values of $C_w^t$ and $V^t$ at the injection phase are

$$C_w^t\big|_{t=0} = 0, \tag{18}$$

$$V^t\big|_{t=0} = \pi r_w^2 (H_w|_{t=0}), \tag{19}$$

where $H_w$ represents the water depth of the wellbore.

In the chase phase, one has

$$C_w^t|_{t=t_{inj}^-} = C_w^t|_{t=t_{inj}^+}, \tag{20}$$

$$V^t|_{t=t_{inj}^+} = \pi r_w^2 \left( H_w|_{t=t_{inj}^+} \right), \tag{21}$$

where the - and + signs in the subscripts of Eqs. (20) - (21) hereinafter represent approaching the limit from left and right
sides of $t_{inj}$, respectively.

## 2.3 Capability of the new SWPP model of this study

Different from the model of Wang et al. (2017), the multi-species reactive transport models are used to describe the
non-linear biogeochemical reactive processes considering wellbore effects not only for groundwater flow but also for solute
concentrations. The new model of this study is an extension of Phanikumar and McGuire (2010) that ignored the wellbore
storage for both groundwater flow and solute transport, and assumed that the aquifer hydraulic diffusivity was infinite. The
Danckwerts condition rather than the Robin condition is applied at the well screen in the rest phase of this study. Therefore,
the new model is more powerful in describing an arbitrary number of species and user-defined reaction rate expressions,
including Monod/Michaelis-Menten kinetics.

## 3. Numerical solution of the SWPP test

In this study, we will use a finite-difference method to solve the model of the SWPP test, where the finite-difference
scheme of the groundwater flow is the same with Wang et al. (2017), and the scheme of the transport governing equation
(ADE) is similar to the model of Phanikumar and McGuire (2010). However, the flow velocity used in the advective term of
ADE is computed by solving the model of groundwater flow rather than directly using Eq. (2), which was employed by
Phanikumar and McGuire (2010). The code of this study is free of charge upon request from the authors.

To minimize numerical errors and to increase computational efficiency, we employ a non-uniform grid system for
simulations (Wang et al., 2014), which is:

$$r_i = \frac{r_{i-1/2}+r_{i+1/2}}{2}, i = 1, 2, 3, \cdots, N_r, \tag{22}$$

where $N_r$ represents the number of nodes in discretization of the spatial domain $[r_w, r_e]$, $r_w$ and $r_e$ respectively represent the
distances of inner and outer boundary nodes; $r_i$ is the radial distance of node; $r_{i+1/2}$ is calculated as follows

$$log_{10}(r_{i+1/2}) = log_{10}(r_w) + i \left[ \frac{log_{10}(r_e)-log_{10}(r_w)}{N} \right], i = 0, 1, 2, \cdots, N_r. \tag{23}$$

The value of $r_{i-1/2}$ can be calculated using the similar way. Eqs. (22) - (23) represent a space domain discretized

logarithmically, and the spatial steps are smaller near the wellbore and become progressively greater away from the wellbore.

Similarly, we logarithmically discretize the temporal domain:

$$t_i = \frac{t_{i-1/2} + t_{i+1/2}}{2}, i = 1, 2, 3, \cdots, M, \tag{24}$$

where $M$ represents the number of nodes in discretization of the temporal domain; $t_i$ is the time of node $i$; $t_{i+1/2}$ is calculated as follows in the injection phase

$$log_{10}(t_{i+1/2}) = log_{10}(t_0) + i\left[\frac{log_{10}(t_{inj}) - log_{10}(t_0)}{M}\right], i = 1, 2, 3, \cdots, M. \tag{25}$$

where $t_0$ is a very small positive value representing the first time step, such as $t_0 = 1.0 \times 10^{-7}$ hour.

As for the chase, one has

$$t_{i+1/2} = 10^{log_{10}(t_0) + i\left[\frac{log_{10}(t_{cha}) - log_{10}(t_0)}{M}\right]} + t_{inj}, i = 1, 2, 3, \cdots, M. \tag{26}$$

Similarly, in the rest phase, one has

$$t_{i+1/2} = 10^{log_{10}(t_0) + i\left[\frac{log_{10}(t_{res}) - log_{10}(t_0)}{M}\right]} + t_{inj} + t_{cha}, i = 1, 2, 3, \cdots, M. \tag{27}$$

In the extraction phase, one has

$$t_{i+1/2} = 10^{log_{10}(t_0) + i\left[\frac{log_{10}(t_{ext}) - log_{10}(t_0)}{M}\right]} + t_{inj} + t_{cha} + t_{res}, i = 1, 2, 3, \cdots, M. \tag{28}$$

Before using the new model of this study, it is necessary to evaluate the numerical errors (like artificial oscillation and numerical dispersion) of the solution. Unfortunately, the benchmark analytical solutions of the SWPP test with a finite hydraulic diffusivity are not available up to date. Alternatively, the accuracy of the finite-difference solution could be tested by comparing with the numerical solution of Wang et al. (2017) which was proven to be accurate and robust. Figure 2 shows the comparison of BTCs between the solution of Wang et al. (2017) and of this study, where the parameters used are similar to Figure 3 of Phanikumar and McGuire (2010): $B=8$ m, $r_w=0.052$ m, $\alpha_r=1$ m, $\theta=0.38$ m, $D_0=0$ m$^2$/hour, $t_{inj}= 94.32$ hour, $t_{cha}=0$ hour, $t_{res}=0$ hour, $t_{ext}=405.6$ hour, injection flow rate $Q_{inj}=0.1$ m$^3$/hour, and extraction flow rate $Q_{ext}=-0.11$ m$^3$/hour. It shows a small oscillation in the numerical solutions, which might be caused by the numerical errors.

By comparing the solution of this study with Wang et al. (2017), one may conclude that the solution of this study appears to be accurate and reliable since the mean square error between two solutions is smaller than 0.05 for all cases in Figure 2. In the wellbore ($r = r_w$), the concertation is equal to $C_0^{inj}$, as shown in Figure 2. This is due to the boundary condition of the wellbore, e.g.,

$$C|_{r \to r_w} = C_0^{inj}, 0 < t \le t_{inj}. \tag{29}$$

In the aquifer, the values of BTCs increase with the decreasing distance from the wellbore.

It is also ~~very~~ necessary to test the accuracy of the new models against the numerical software packages. Since the code of the original MODFLOW/MT3DMS package is open source and could be downloaded freely from the website of United State Geological Survey, it is preferred by many modellers and is selected as the base of comparison in this study. Unfortunately, such an open-source MODFLOW/MT3DMS package may create some errors in describe the solute transport in the wellbore-confined aquifer. The errors come from an assumption that the water volume in the wellbore is computed by a product of wellbore cross section and the aquifer thickness, which is incorrect. The actual water volume in the wellbore should be computed by a product of wellbore cross section and the water level in the wellbore (See Supplemental Materials for detailed explanation). Figure 3 shows the comparisons of BTCs between the open-source MODFLOW/MT3DMS ~~model package~~ and new model of this study. The water level of the wellbore is assumed to be equal to the aquifer thickness in the new models for the purpose of comparison although it may be not true, and the other parameters used are the same with ones in Figure 2. Therefore, the agreement between two models demonstrates the accuracy of the new model. Figure 3 show that the concentration in the wellbore is not unit in the injection phase, and this is because the new model considers the wellbore storage for both groundwater flow and solute transport. It is worthwhile to point out that an advanced version of MODFLOW/MT3DMS, namely the MODFLOW-SURFACT includes a fracture-well package (FWL4 and FWL5) to overcome the problems in the original open-source MODFLOW well package. The FWL4 and FWL5 packages calculate the water volume using simulated heads, not aquifer thicknesses (See MODFLOW-SURFACT manual, Vol I, Section 3.2, Eq. 24 for details). FEFLOW also has a similar package, referred to as discrete-feature to simulate a pumping/extraction well, if one chooses to do so. Additionally, with a FEFLOW model, the model mesh can be highly discretized to accurately represent well dimensions using a subset of elements (in centers). The modeller can assign a porosity of unit for those elements representing the wells, rather than assuming the same porosity of the surrounding materials. In the future, we will conduct a comprehensive comparative investigation of the method proposed in this study and those of MODFLOW-SURFACT and FEFLOW for understanding the effects of well mixing and wellbore storage for both flow and transport processes involving an aquifer-well system.

## 4. Discussions: Effect of wellbore storage on the SWPP test under transient flow field

Revisiting the assumptions used in previous studies as mentioned in Sections 2.1 and 2.2, one may find that the flow field and the wellbore storage are key factors for the SWPP test. This is not surprising, since the flow velocity is not only included in the advective term, but also in the dispersive term. The wellbore storage which is dependent on the volume of pre-test water in the wellbore may influence the concentration of the solute injected into the wellbore. As the influence of the

hydraulic diffusivity solute transport in the SWPP test has been investigated in Wang et al. (2017), in this section, we mainly investigate the influence the wellbore storage on the reactive transport in the SWPP test in the transient flow field.

The variation of the transient flow field is mainly controlled by the hydraulic diffusivity of the aquifer and the wellbore storage. In the following discussion, we choose three representative types of porous media to test the influence of the hydraulic diffusivity on the results of the SWPP test, including fine sand, medium sand, and coarse sand. According to Domenico and Schwartz (1990) and Batu (1998), one could obtain the values of the hydraulic diffusivity for above mentioned three types of media: $4.17 \times 10$ m$^2$/hour (with $K_r = 4.17 \times 10^{-3}$ m/hour and $S_s = 1.0 \times 10^{-4}$ m$^{-1}$) for the fine sand, $4.17 \times 10^2$ m$^2$/hour (with $K_r = 4.17 \times 10^{-2}$ m/hour and $S_s = 1.0 \times 10^{-4}$ m$^{-1}$) for the medium sand, and $4.17 \times 10^4$ m$^2$/hour (with $K_r = 4.17 \times 10^{-1}$ m/hour and $S_s = 1.0 \times 10^{-5}$ m$^{-1}$) for the coarse sand. Generally, the hydraulic diffusivity of the aquifer correlates to the grain size of the media, and the value is smaller for the smaller grain size, e.g., fine sand.

The parameters related to the solute transport mainly come from the studies of Phanikumar and McGuire (2010), who interpreted the field experimental data of the SWPP test conducted by McGuire et al. (2002). Except for parameters specifically mentioned otherwise, the default values used in the following section are $C_0^{inj}$=100 mg/L, $C_0^{cha}$=10 mg/L, $B$=0.1 m, $r_w$=0.0125 m, $\alpha_r$=0.01 m, $\theta$ =0.33, $D_0$=0 m$^2$/hour, $t_{inj}$=0.6 hour, $t_{cha}$=0.067 hour, $t_{res}$=0.0333 hour, $t_{ext}$=3.6 hour, $Q_{inj}$=0.0333 m$^3$/hour, $Q_{cha}$=0.0255 m$^3$/hour, and $Q_{ext}$=0.0333 m$^3$/hour, which can be found in Figure 5 of Phanikumar and McGuire (2010).

## 4.1 The rest phase

Figures 4A and 4B show the comparison of BTCs between the Robin and Danckwerts conditions at the wellbore for different porous media, where $C_0^{cha}$=10 mg/L in Figure 4A and $C_0^{cha}$=0.0 mg/L in Figure 4B. For the purpose of comparison, the boundary conditions at the wellbore in the injection and chase phases are still described by Eqs. (5) - (6).

Figure 4A shows that the difference of BTCs between two boundary conditions is significant at the early stage of the extraction phase when $C_0^{cha}$=10 mg /L, and BTCs of the Danckwerts condition are above BTCs of the Robin condition. With time going, such a difference becomes negligible. As for the curves of the Robin condition, the solute concentration in the wellbore is 0 in the chase phase, correspondingly, the concentration starts from 0 at the early stage of the extraction phase. Actually, the solute concentration in the wellbore may be not 0 in the rest phase due to the wellbore storage and finite hydraulic diffusivity when $C_0^{cha}$=10 mg /L. Another interesting observation is that the properties of the porous media could also influence the difference of BTCs between two boundary conditions. Obviously, a smaller hydraulic diffusivity would result in a larger difference between them, e.g., such a difference is greater for the fine sand aquifer.

Figure 4B shows the comparison of BTCs for different boundary conditions in the wellbore when $C_0^{cha}$=0.0, and one could find that the difference of BTCs between the Robin and Danckwerts conditions is negligible, which implies that the Robin condition performs well when $C_0^{cha}$=0.0, while not for the case $C_0^{cha}$ ≠0.0.

## 4.2 The injection and chase phases

Figure 5 shows the comparison of BTCs in the wellbore for different boundary conditions and different porous media. The parameters used in this case are the same as ones in Section 4.1. The initial head is 1 m. The boundary condition of the wellbore in the rest phase is described by the Danckwerts condition.

Two interesting observations can be seen from this figure. Firstly, the difference of BTCs between the two boundary conditions at the wellbore is obvious, and such a difference is larger for the medium sand than for the coarse sand, implying that it increases with the decreasing hydraulic diffusivity. Secondly, the values of BTCs obtained from Eqs. (16) - (17) are greater at the early stage of the extraction phase, while the peak values of BTC are smaller. In another word, the model of Eqs. (5) - (6) may underestimate the concentration in the early stage of the extraction phase, while overestimate the peak values of BTCs.

These observations can be explained as follows. The model of Eqs. (5) - (6) assumes that the volume of water in the wellbore is negligible, and the concentration in the wellbore is close to 10.0 mg/L in the rest phase, due to $C_0^{cha}$=10.0 mg/L. As for the model of Eqs. (16) - (17), the volume of water in the wellbore is non-negligible and could dilute the concentration in the injection phase; namely, the solute concentration in the wellbore could not immediately rises to $C_0^{inj}$ at the early stage of the injection phase, thus resulting in smaller peak values of BTCs. Similarly, the concentration in the wellbore could not immediately reduce to $C_0^{cha}$ at the early stage of the chase phase, which makes the concentration larger at the early stage of the extraction phase based on the model of Eqs. (16) - (17).

## 5. Uniqueness of estimated parameters

Physical and chemical parameters are important in predicting the contaminate transport in the aquifer, and the values of these parameters are generally estimated by best fitting the observed BTCs in the SWPP test using a simplified model, ignoring a number of relevant factors such as the influences of the flow field and the wellbore storage. The discussions in Section 4 demonstrate that the negligence of such factors on reactive transport might cause errors and invalidate the whole parameter estimation exercises. Beside porosity, dispersivity, and reaction rates, the new model of this study appears to be useful for estimating the values of hydraulic conductivity and specific storage by best fitting the observed BTCs in the SWPP test. For instance, the values could be determined by minimizing the sum of absolute differences between the observed and calculated BTCs in the wellbore

$$\mathcal{F} = \sum_{i=1}^{o} \left| C_{CAL}\left(K_r, S_s, \alpha_r, \theta, \lambda_j, t, r_w\right)\big|_{t=t_i} - C_{OBS}(t, r_w)\big|_{t=t_i} \right|, \tag{30}$$

where $C_{CAL}\left(K_r, S_s, \alpha_r, \lambda_j, t, r_w\right)\big|_{t=t_i}$ and $C_{OBS}(t, r_w)|_{t=t_i}$ represent the concentrations calculated by the new model of the SWPP test and the observed concentrations at $t = t_i$, respectively; $o$ is the number of observed data.

Although the number of observation points is usually much greater than the number of parameters needed to be estimated, one may still wonder if Eq. (30) is practically reliable for estimation of multiple parameters simultaneously. To answer this question, two approaches are employed in the following: Sensitivity analysis and uniqueness analysis. The sensitivity analysis is used to check whether the solution is sensitive to the parameters or not, while the uniqueness analysis is to check if the multiple input parameter values could map to the same output results.

## 5.1 Sensitivity analysis

McCuen (1985) proposed a sensitivity model of a dependent variable, which was normalized as (Kabala, 2001; Yang and Yeh, 2009):

$$SC_{i,j} = I_j \frac{\partial c_i}{\partial I_j}, \tag{31}$$

where $SC_{i,j}$ is the sensitivity coefficient of the $j^{th}$ parameter $I_j$ at the $i^{th}$ time; $C_i$ is the concentration at the $i^{th}$ time. In this study, the differentiation of Eq. (31) will be approximated by a finite-difference scheme:

$$SC_{i,j} = I_j \frac{c_i(I_j + \Delta I_j) - c_i(I_j)}{\Delta I_j}, \tag{32}$$

where $\Delta I_j$ is a small increment.

From the mathematic models of the groundwater flow, one may find both hydraulic conductivity and specific storage could affect the flow field. Since greater hydraulic conductivity or smaller specific storage could shorten the time approaching the steady state, we will employ the hydraulic diffusivity for the sensitivity analysis, which is the ratio of the two parameters. Figure 6 shows the sensitivity of the hydraulic diffusivity on BTCs, and one may find that it is not sensitive to the hydraulic diffusivity when the values of hydraulic diffusivity are sufficiently large. This might be because the time approaching the steady state is very short when the hydraulic diffusivity values are sufficiently large (for instance, greater than $4.17 \times 10^2$ m$^2$/hour), and the influence of the transient flow could be ignored. Therefore, the steady-state assumption could be used to approximate the flow field in the SWPP test when the hydraulic diffusivity is greater than $4.17 \times 10^2$ m$^2$/hour. Otherwise, the steady-state assumption is not recommended. Figure 7 shows that the BTCs in the wellbore are sensitive to both dispersivity and porosity.

## 5.2 Uniqueness analysis of physical parameters

Beside the sensitivity analysis, the uniqueness analysis is also important for the parameter estimation, which is used to check if there exist two or more sets of parameters for the same BTCs. Similar to the treatment in previous studies, we firstly use the transient model of this study to reproduce BTCs based on a set of given input parameters, and then estimate the values of parameters by best fitting such BTCs. If the values of the input parameters are different from the estimated

parameter when the fitness is very good, one could conclude that the solution is not unique and the parameters estimated from Eq. (30) may not be reliable.

There are four physical parameters in the new model of this study, including hydraulic conductivity, specific storage, dispersivity, and porosity, and one chemical parameter (reaction rate). Wang et al. (2017) investigated the uniqueness of solutions for the flow field, and the results showed that BTCs of the SWPP test were not unique for the flow-related parameters. For instance, BTCs with a steady state flow field were almost the same with BTCs with a transient flow field, as shown in Figures 10 and 11 of Wang et al. (2017). It implies that one may not inversely determine the hydraulic parameters of flow field only by best fitting observed BTCs in the wellbore, and additional aquifer tests are required to supplement the SWPP test to determine the flow-related parameters. However, Wang et al. (2017) did not investigate the uniqueness of porosity and dispersion when the hydraulic parameters were given, which will be discussed in this study.

Figure 8 shows comparison of BTCs for different dispersivities and porosities but for the same hydraulic parameters, and one could see that the curves of $\alpha_r = 0.01$m and $\theta = 0.33$ are almost the same with the curves of $\alpha_r = 0.006$ m and $\theta = 0.9$. Therefore, Eq. (30) may not be used to determine $\alpha_r$ and $\theta$ simultaneously. Fortunately, the porosity could be measured in the laboratory from core samples or determined by the SWPP test with drift flow (Hall et al., 1991; Paradis et al., 2018). When the values of $K_r, S_s,$ and $\theta$ are given, the dispersivity could be determined uniquely by Eq. (30).

In summary, it seems impossible to determine all parameters ($K_r, S_s, \alpha_r,$ and $\theta$) simultaneously by only best fitting the observed BTCs in the wellbore of the SWPP test using Eq. (30). Therefore, before determining the parameters related to the solute transport ($\alpha_r$ and $\theta$), the hydraulic parameters ($K_r$ and $S_s$) needed to be estimated by supplementary aquifer tests, or by best-fitting the pressure data measured during the SWPP test, e.g. the pumping phase. The value of $\alpha_r$ could be determined by Eq. (30) when the porosity is given.

## 5.3 Chemical parameter estimation

The models estimating the reaction rate are based on several assumptions in previous studies, e.g. Eqs. (8) - (9) as demonstrated in Section 2.1. To test the applicability of those equations, we will use the model of this study to reproduce the data of $\ln(C_{rec}/C_{tra}) \sim t^*$ based on a set of given parameters, and then using Eqs. (8) - (9) (which is based on an infinite hydraulic diffusivity presumption) to estimate $\lambda$ (denoted as $\tilde{\lambda}$) by best fitting $\ln(C_{rec}/C_{tra}) \sim t^*$. Two species involved in this case are Cl[-1] and SO₄[-2], in which $C_{tra}$ and $C_{rec}$ represent the concentrations of Cl[-1] and SO₄[-2], respectively. Figure 9 shows the fitness of the simulated $\ln(C_{rec}/C_{tra}) \sim t^*$ in the wellbore using a linear function, with the detailed information is shown in Table 1. Two sets of $\lambda$ are employed in the discussions for the reactant, e.g. $\lambda = 0.1$ hour[-1] and 0.2 hour[-1]. One may conclude that the simplified models of Eqs. (8) - (9) with an infinite hydraulic diffusivity perform well in the estimation of $\lambda$ for reactive transport under the finite hydraulic diffusivity condition.

This simplified model of Eq. (9) has been widely used to estimate $\lambda$, due to the advantages that $\lambda$ could be determined directly by best fitting the observed $\ln(C_{rec}/C_{tra})\sim t^*$, without the knowledge of the aquifer properties, such as porosity, dispersivity, hydraulic diffusivity. However, this model is proposed based on the first-order reaction assumption, which is a linear function as shown in Eq. (7). Whether this model works for non-linear reactions or not is still unknown, and will be investigated in the following section.

Assuming that the extraction time since the rest phase ended could be divided into $N$-1 segments, Phanikumar and McGuire (2010) employed the Heaviside unit step function to describe a type of non-linear biogeochemical reaction:

$$\frac{\partial C}{\partial t} = -\sum_{j=1}^{N-1}[\mathcal{H}(t-t_j^*) - \mathcal{H}(t-t_{j+1}^*)]\lambda_j C_i^{n_j}, \tag{33}$$

where $\lambda_j$ is the reaction constant in the temporal segment $j$, and the Heaviside step function $\mathcal{H}(\cdot)$ is:

$$\mathcal{H}(t-t_j^*) - \mathcal{H}(t-t_{j+1}^*) = \begin{cases} 0 & if\ t < t_j^* \\ 1 & if\ t_j^* < t < t_{j+1}^* \\ 0 & if\ t_{j+1}^* < t \end{cases}. \tag{34}$$

Eq. (33) is a series of piece-wise linear ($n_j$ =1) or non-linear ($n_j \neq$ 1) functions, which are an extension of Eq. (7).

To test the influence of the hydraulic diffusivity on the accuracy of this model in estimating $\lambda_j$ for the non-linear reactions, the model of this study is used to reproduce the data of $\ln(C_{rec}/C_{tra})\sim t^*$ with a set of specific $\lambda_j$, $n_j$ and $t_j^*$ for three types of porous media. Figures 10 and 11 represent the computed $\ln(C_{rec}/C_{tra})\sim t^*$ based on the model of the chemical reactions described by the piece-wise linear and the non-linear functions, respectively. The values of $\lambda_j$ and $t_j^*$ of Figure 10 are obtained by best fitting the observation data using a piecewise linear function (e.g., $n_j$ =1) proposed by Phanikumar and McGuire (2010). The circle represents the experiments data observed by McGuire et al. (2002). The parameters related to the chemical reactions in Figure 11 are from Phanikumar and McGuire (2010) by best fitting the observation data using a nonlinear function: $\lambda_j$ =0.25, $n_j$ =0.25, $N=j$=1. Comparing Figures 10 and 11, one may find that the influence of the hydraulic diffusivity on the computed $\ln(C_{rec}/C_{tra})\sim t^*$ is negligible for the chemical reaction described by the piecewise linear function, which is similar to the first-order reaction as shown in Figure 9. However, the influence of the hydraulic diffusivity on the relationship of $\ln(C_{rec}/C_{tra})\sim t^*$ cannot be ignored if one attempts to use nonlinear functions to describe such a chemical reaction. The difference between the curves of different porous media is obvious in Figure 11. The agreement between the observed and computed data is satisfactory for the medium and coarse sands, but not for the fine sand in Figure 11. This is because the hydraulic diffusivity values of the medium and coarse sands are larger than that of the fine sand, thus are close to the assumption of an infinite hydraulic diffusivity used in Phanikumar and McGuire (2010).

Therefore, one may conclude that $\lambda_j$, $n_j$ and $t_j^*$ could be determined by directly best fitting the observed $\ln(C_{rec}/C_{tra})\sim t^*$ when the non-linear reactions are described by the piece-wise linear functions, in a similar way to estimate the

linear reaction rate by Eq. (7). However, such an approach may not work if one attempts to use non-linear functions to describe such reactions.

## 6. Field applications

To test the model of this study, the field data of a SWPP test conducted in a single well by McGuire et al. (2002) will be employed. In this test, the prepared solution contains $Na_2SO_4$ (as a reactant) and NaCl (as a conservative tracer). The reactant and the tracer were well mixed and then injected into a targeted aquifer.

### 6.1 Revisit of previous model

Phanikumar and McGuire (2010) interpreted such data using a model containing several assumptions mentioned in Section 2.1. The parameters used in their model were $B$=0.1 m, $r_w$=0.0125 m, $\alpha_r$=0.001 m, $\theta$ =0.33 m, $D_0$=0 m$^2$/hour, $t_{inj}$=0.6 hour, $t_{cha}$=0.067 hour, $t_{res}$=0.0333 hour, $t_{ext}$=3.6 hour, $Q_{inj}$=0.0333 m$^3$/hour, $Q_{cha}$=0.0255 m$^3$/hour, and $Q_{ext}$=-0.011 m$^3$/hour. The concentrations of NaCl were $C_0^{inj}$=100 mg/L in the injection phase and $C_0^{cha}$=10 mg/L in the chase phase. As for the reactant of $Na_2SO_4$, the concentrations were $C_0^{inj}$=20 mg/L and $C_0^{cha}$=2 mg/L.

To demonstrate the importance of the wellbore storage of the solute transport, which was ignored in Wang et al. (2017), the observed and computed BTCs are compared based on the estimated parameters in Phanikumar and McGuire (2010), as shown in Figures 12A and 12B. The computed BTCs in Figures 12A and 12B are located $r$= $r_w$+0.15 m and $r$= $r_w$, respectively. The legend of "PPTEST" represents the solution of Phanikumar and McGuire (2010), and the others are produced by the new model ignoring the wellbore storage effect on the solute transport.

The results showed that the fitness between the observed BTCs in the wellbore and computed BTCs by "PPTEST" was very well, as shown in Figure 12A of this study, or Figure 5 of Phanikumar and McGuire (2010). However, by carefully checking the report of Phanikumar (2010), we found that the computed BTCs were at a radial distance of 0.15 m from the wellbore, rather than at the wellbore itself in Phanikumar (2010). They did not provide a convincing argument why to choose BTCs in the aquifer to represent BTCs in the wellbore, and thus the use of "0.15 m" in their analysis appears to be an artefact, rather than being physically based. Figure 12B shows the comparison of the computed and observed BTCs in the wellbore for different hydraulic diffusivities. Obviously, the new model ignoring the wellbore storage of the solute transport could not be used to interpret experimental data, since the computed BTCs are zero at the early stage of the extraction phase.

From Figures 12A and 12B, several interesting observations could be made. Firstly, the difference of BTCs among different porous media is obvious. BTCs of the coarse sand aquifer are close to the solution of "PPTEST", as shown in Figure 12A. This is because the hydraulic diffusivity of the coarse sand aquifer is the largest, which is close to the assumption used in "PPTEST" that hydraulic diffusivity is infinity. Secondly, the wellbore concentration is 10 mg/L at the

early stage of the extraction phase for Cl⁻. This is mainly due to the chosen boundary condition at the well screen, which has been discussed in details in Section 4.1. Thirdly, the peak values of BTCs increase with the decreasing hydraulic diffusivity, and the arrival times of peak values increase with the decreasing hydraulic diffusivity. Such an observation is also found in Figures 4A and 4B. Fourthly, the configuration of BTCs in the aquifer (at $r= r_w+0.15$ m) computed by the model of this study shows that the concentration firstly decreases with time, then increases with time, as shown in Figure 12A. This observation could be explained by the corresponding flow field, as shown in Figure 13. Looking at the flow velocity in the aquifer at $r= r_w+0.15$ m, one may find that the flow direction is still outward from the wellbore in the early stage of the extraction phase, due to the finite hydraulic diffusivity. The outward flow will persist for a finite period of time, depending on the value of the hydraulic diffusivity, and then reverse its direction to flow towards the wellbore for the rest of the extraction phase. This feature is very different from the results with an infinite hydraulic diffusivity assumption, in which the flow direction is always towards the wellbore for the entire extraction phase.

## 6.2 Fitness of this study

We try to use the new model to interpret BTCs of the SWPP test, considering a finite hydraulic diffusivity, a finite wellbore storage, and new boundary conditions of the wellbore at the injection, chase and rest phases. Assuming the initial head of the flow field is 1 m. In a trial and error process of best fitting the observed BTC data, we only estimate parameters of $K_r$, $S_s$ and $\alpha_r$, while the other parameters are the same with those used for producing Figure 5 of Phanikumar and McGuire (2010). Figure 14 demonstrates the fitness of the observed BTC data in the wellbore when $K_r = 1.0$ m/hour, $S_s = 1.0\times10^{-5}$ m⁻¹ and $\alpha_r=0.015$ m. Since the hydraulic diffusivity of this case greater than the hydraulic diffusivity of medium sand ($4.17\times10^2$ m²/hour), the influence of the flow field could be negligible. In this study, we mainly estimated the value of dispersivity, where the porosity is fixed and comes from the reference of Phanikumar and McGuire (2010). Therefore, the dispersivity is uniquely determined.

## 6 Summary and conclusions

A complete SWPP test includes injection, chase, rest and extraction phases, where the second and third phases are not necessary but are recommended to increase the duration of reaction. Due to the complex mechanics of biogeochemical reactions, aquifer properties, and so on, previous mathematic or numerical models contain some assumptions which may over simplify the actual physics, for instance, the hydraulic diffusivity of the aquifer is infinite. The Robin or the third-type boundary condition was often used in previous studies at the well screen in the injection, chase and rest phases by ignoring the mixing effect of the volume of water in the wellbore (namely, wellbore storage). In this study, we presented a multi-species reactive SWPP model considering the wellbore storage for both groundwater flow and solute transport, and a finite aquifer hydraulic diffusivity. The models of wellbore storage for both solute transports are derived based on the mass balance. The Danckwerts boundary condition instead of the Robin condition is employed for solute transport across the well

screen in the rest phase. The robustness of the new model is tested by the field data. Meanwhile, the sensitivity analysis and uniqueness analysis of BTCs in wellbore are conducted. The following conclusion can be made from this study:

(1) The influence of wellbore storage for the solute transport increases with the decreasing hydraulic diffusivity in the injection and chase phases, and the model of Eqs. (16) - (17) underestimates the concentration in the early stage of the injection phase, while overestimates the peak values of BTCs.

(2) The values of $\lambda_j$, $n_j$ and $t_j^*$ could be determined by directly best fitting the observed $\ln(C_{rec}/C_{tra}) \sim t^*$ when the non-linear reactions are described by the piece-wise linear functions, while such an approach may not work if one attempts to use non-linear functions to describe such non-linear reactions.

(3) The Robin condition used to describe the wellbore flux in the rest phase works well only when the chase concentration is zero and the prepared solution in the injection phase is completely pushed out of the borehole into the aquifer, while the Danckwerts boundary condition performs betters even when the chase concentration is not zero.

(4) In the extraction phase, the peak values of BTCs increase with the decreasing hydraulic diffusivity, and the arrival time of the peak value becomes shorter when the hydraulic diffusivity is smaller.

(5) It seems impossible to determine all parameters simultaneously by only best fitting the observed BTCs in the wellbore of the SWPP test using Eq. (30). The hydraulic parameters needed to be estimated by supplementary aquifer tests before determining the parameters related to the solute transport. The value of $\alpha_r$ could be determined by Eq. (30) when the porosity is given.

**Acknowledgments**

This research was partially supported by Program of the Postdoctoral Science Foundation (Nos. 2014M560635 and 2015T80853), Natural Science Foundation of China (No.41502229), Innovative Research Groups of the National Nature Science Foundation of China (No. 41521001), and the National Basic Research Program of China Science Research of Central Colleges, China University of Geosciences (Wuhan) (No. CUGL160407). We sincerely thank two anonymous reviewers for their constructive comments which help us improve the quality of- this manuscript.

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

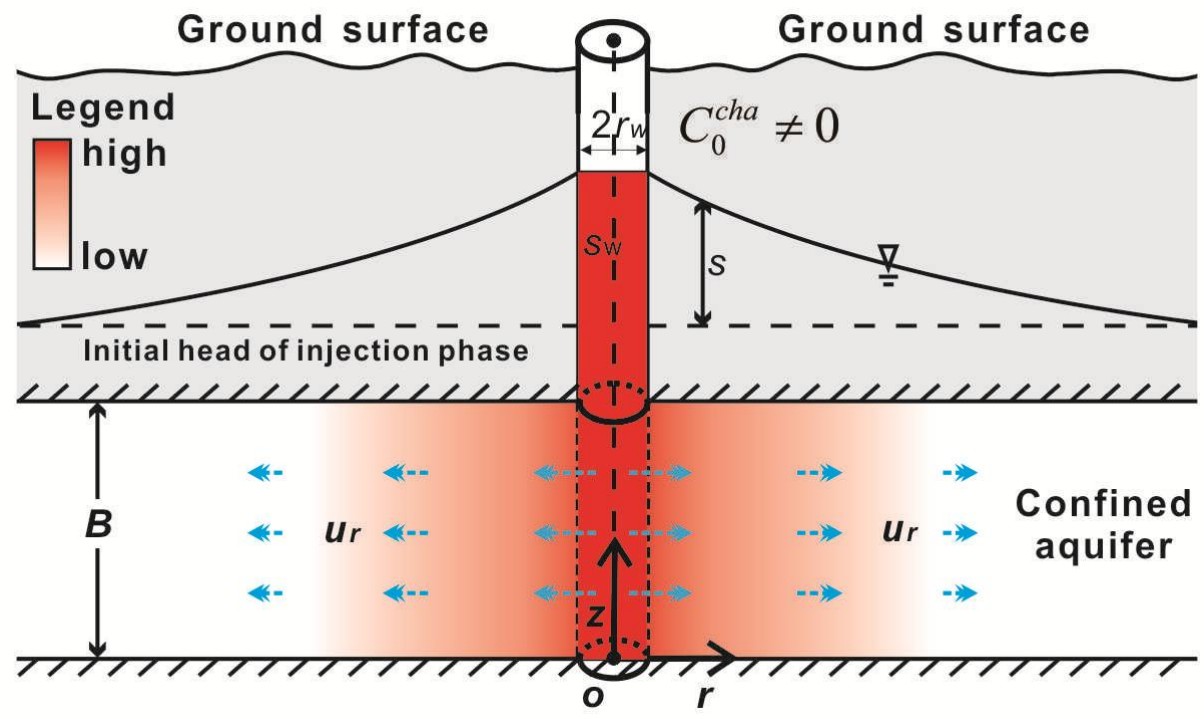

**Figure 1.** The schematic diagram of the SWPP test at the beginning of the rest phase when the chase concentration is not 0.

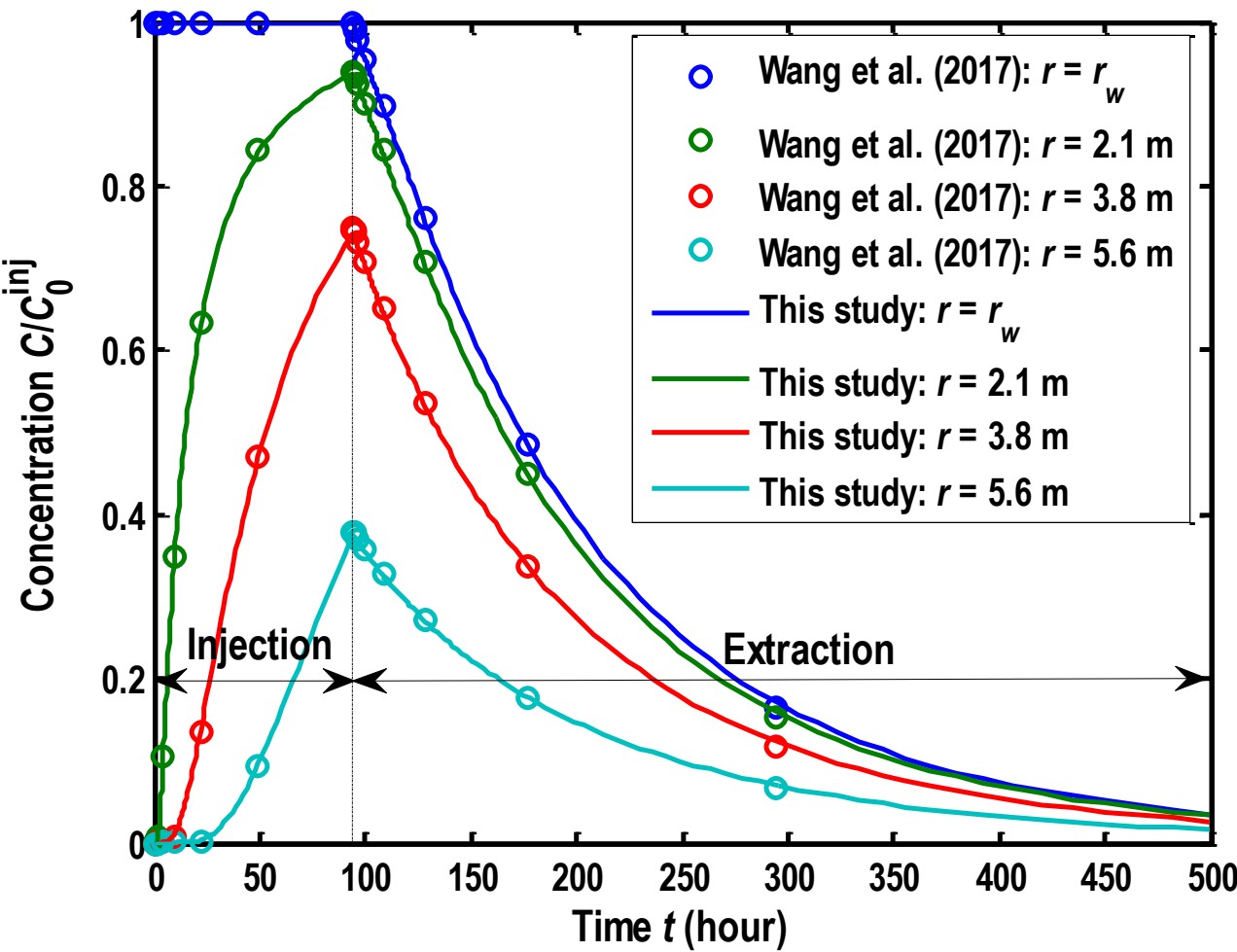

**Figure 2.** Comparison of BTCs between the solutions of Wang et al. (2017) and of this study, where $C_0^{inj}$ represents the concentration of the prepared solute in the injection phase.

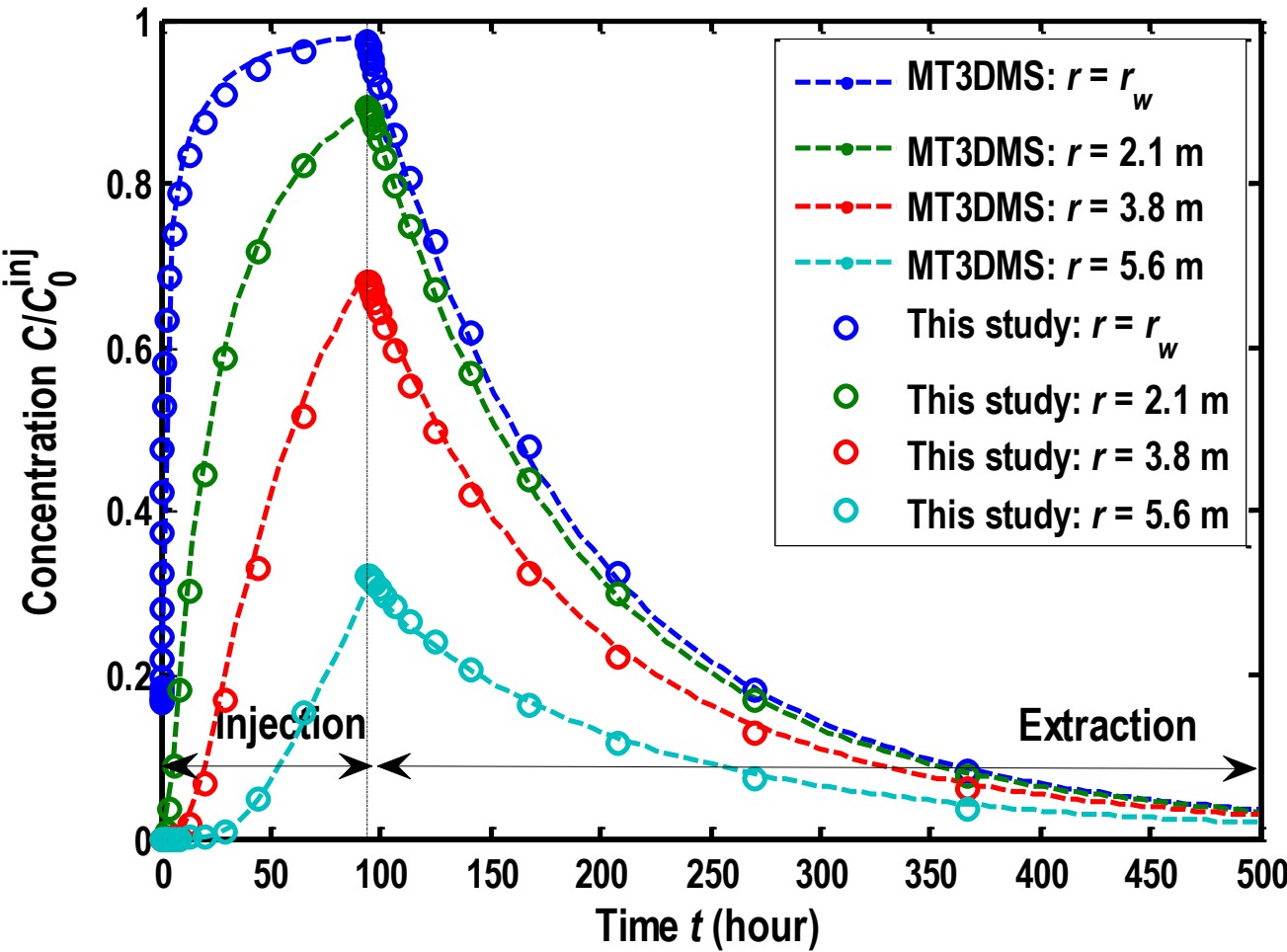

**Figure 3.** Comparison of BTCs between the solutions of MODFLOW/MT3DMS and of this study.

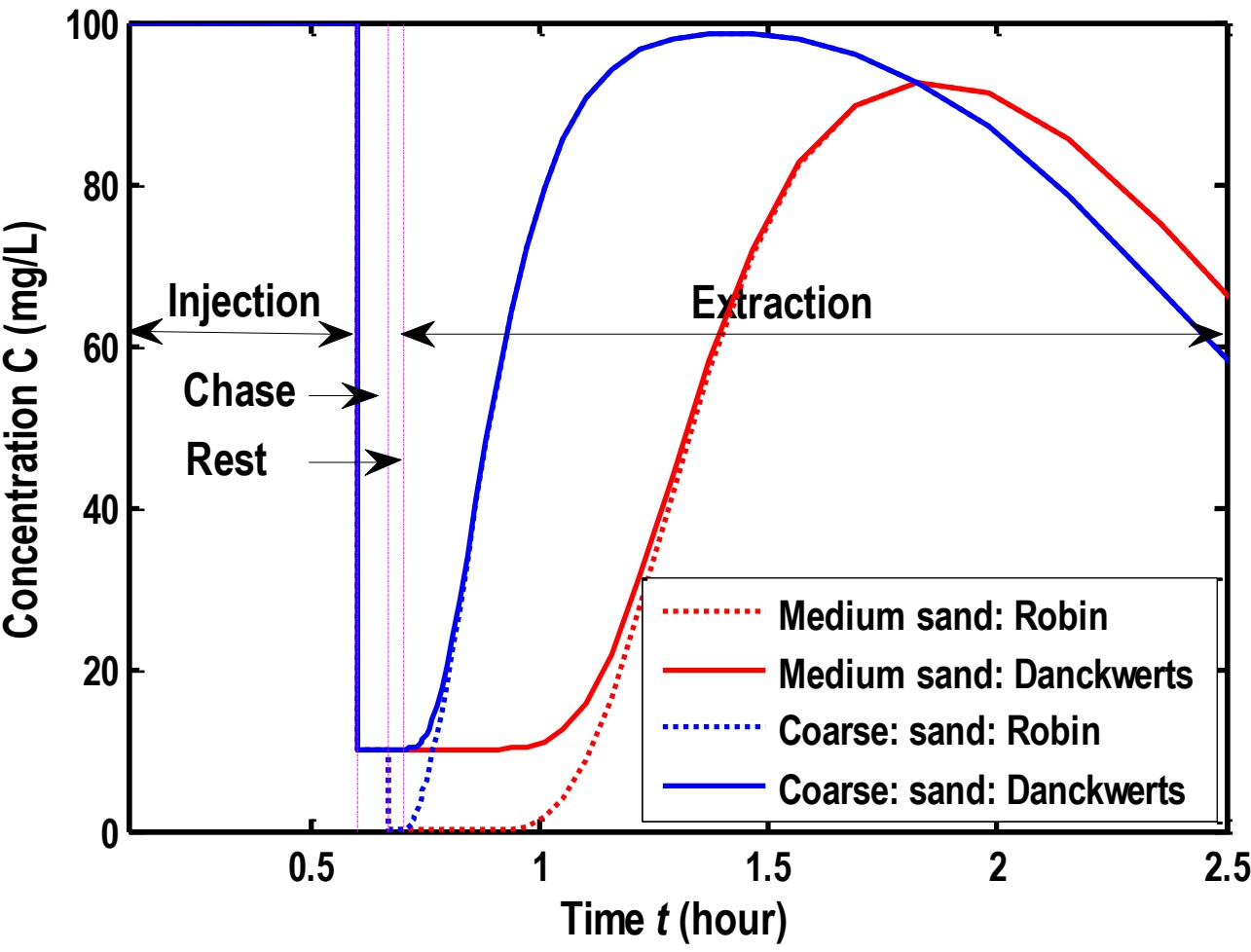

**A**

**Figure 4.** Comparison of BTCs in the wellbore between the Robin and Danckwerts conditions: A. $C_0^{cha}$ =10.0 mg/L, B. $C_0^{cha} = 0$.

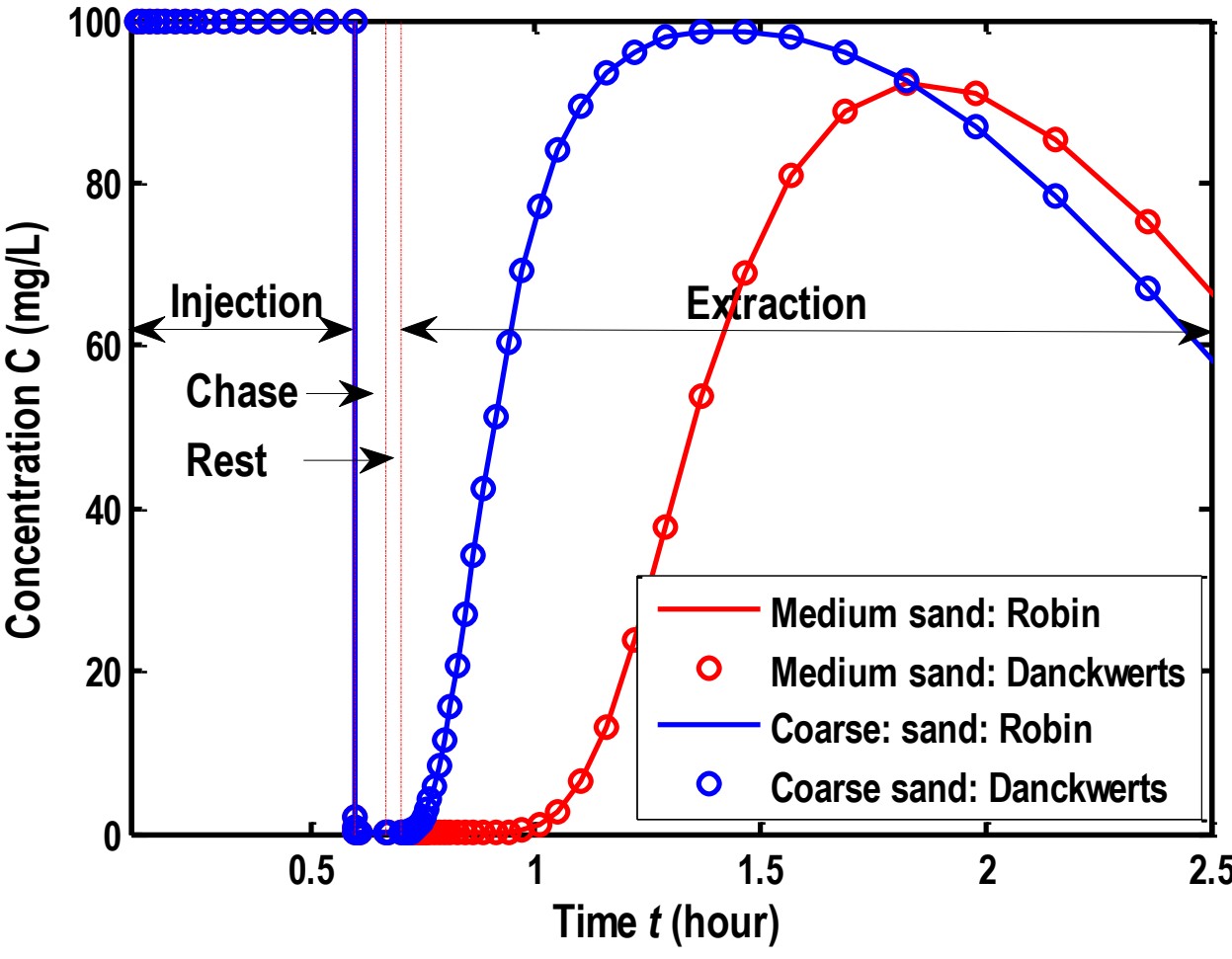

**B**

**Figure 4.** Comparison of BTCs in the wellbore between the Robin and Danckwerts conditions: A. $C_0^{cha}$ =10.0 mg/L, B.

$C_0^{cha} = 0$.

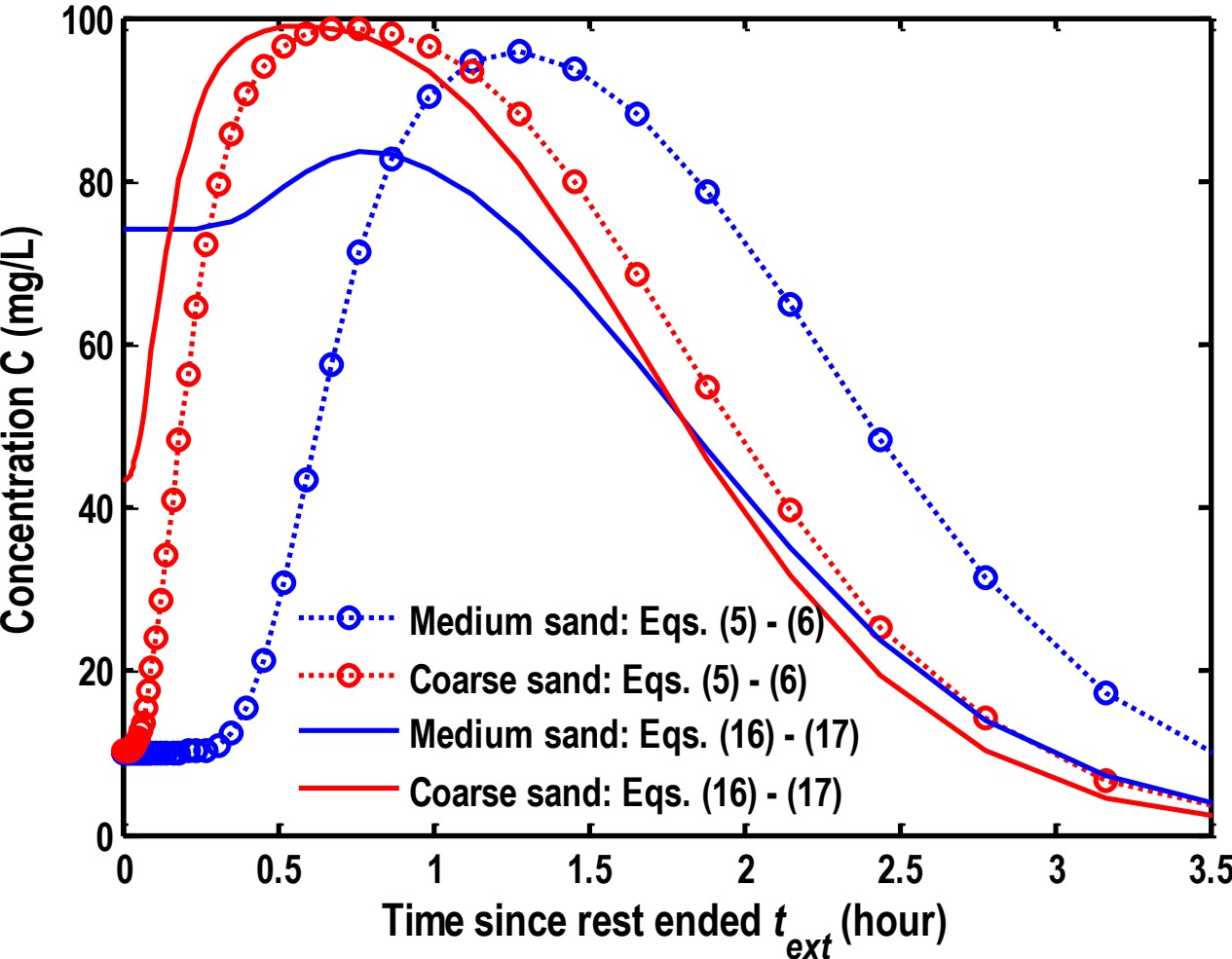

**Figure 5.** The BTCs in the wellbore for the different boundary conditions at the wellbore in the injection and chase phases.

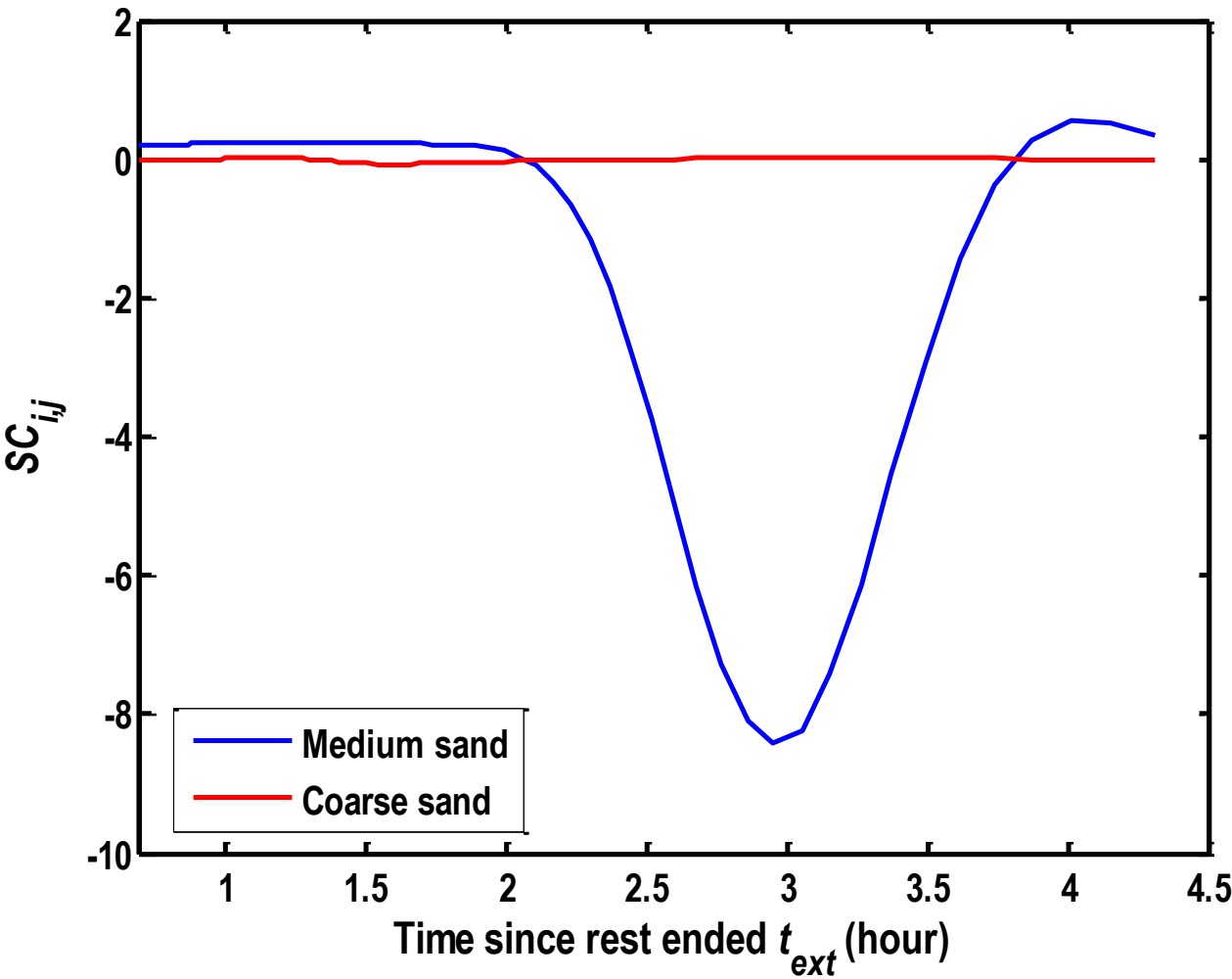

**Figure 6.** Sensitivity analysis of the hydraulic diffusivity on BTCs in the extraction phase

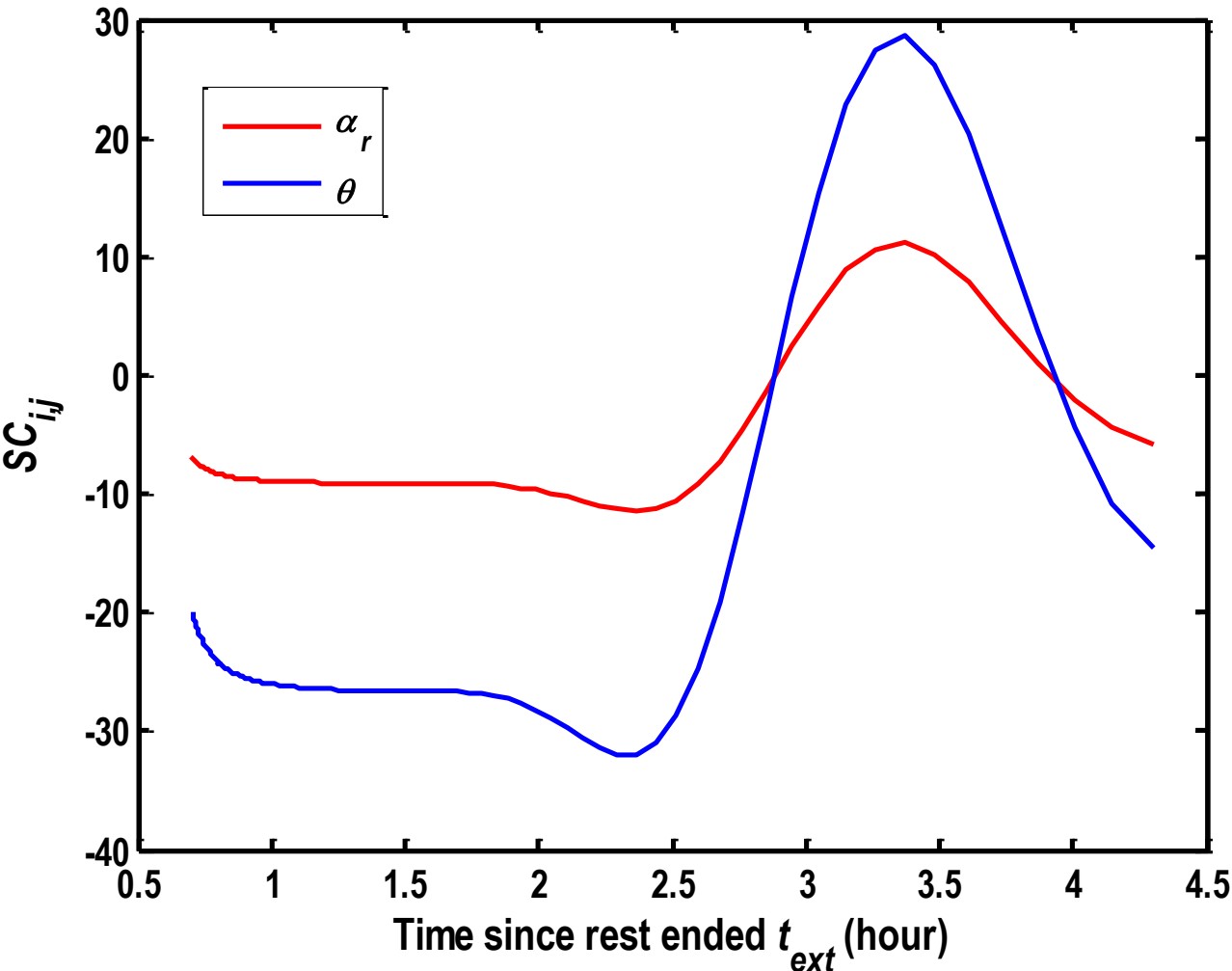

**Figure 7.** Sensitivity analysis of dispersivity and porosity on BTCs in the extraction phase

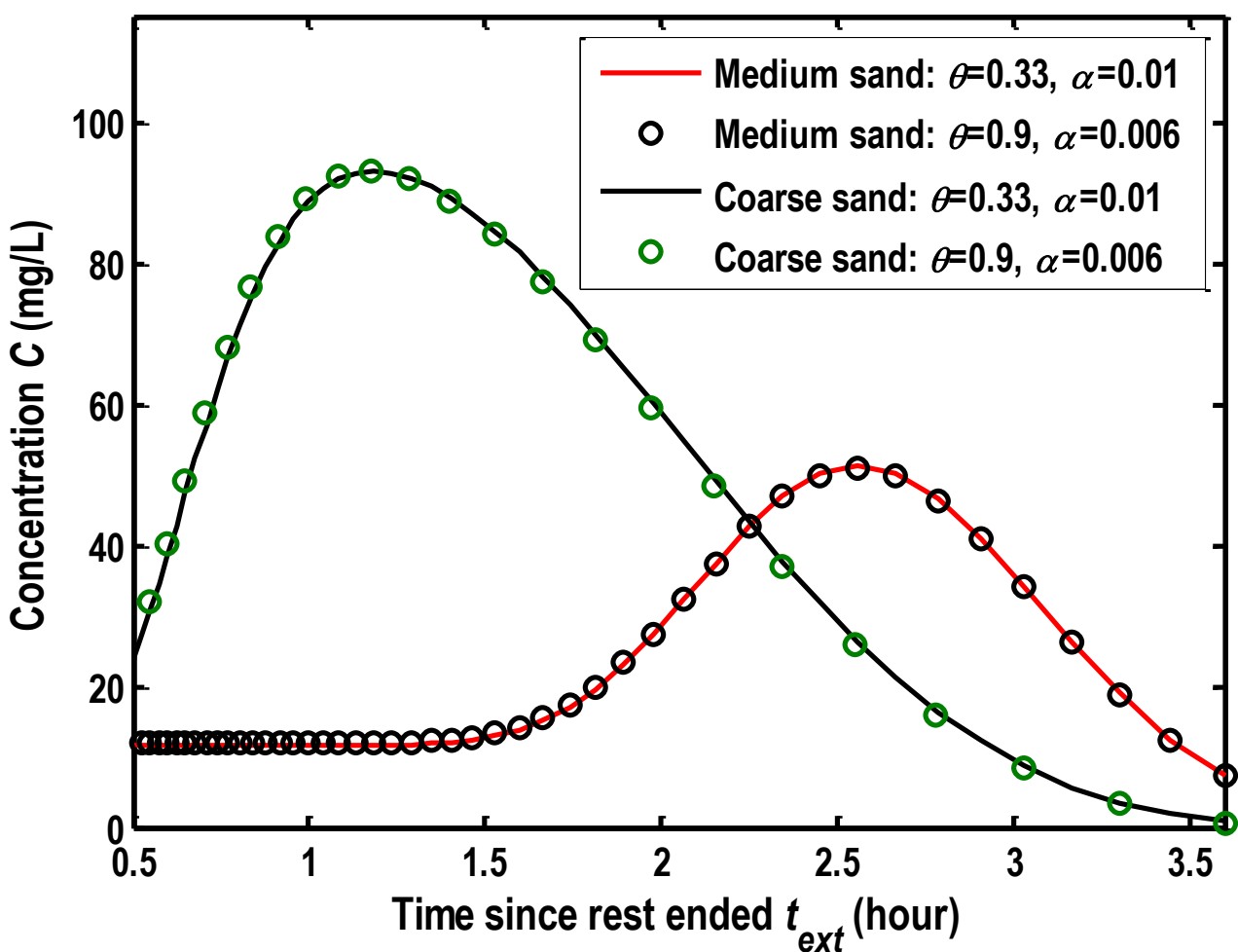

**Figure 8.** Comparison of BTCs for different dispersivities and porosities but for the same hydraulic parameters.

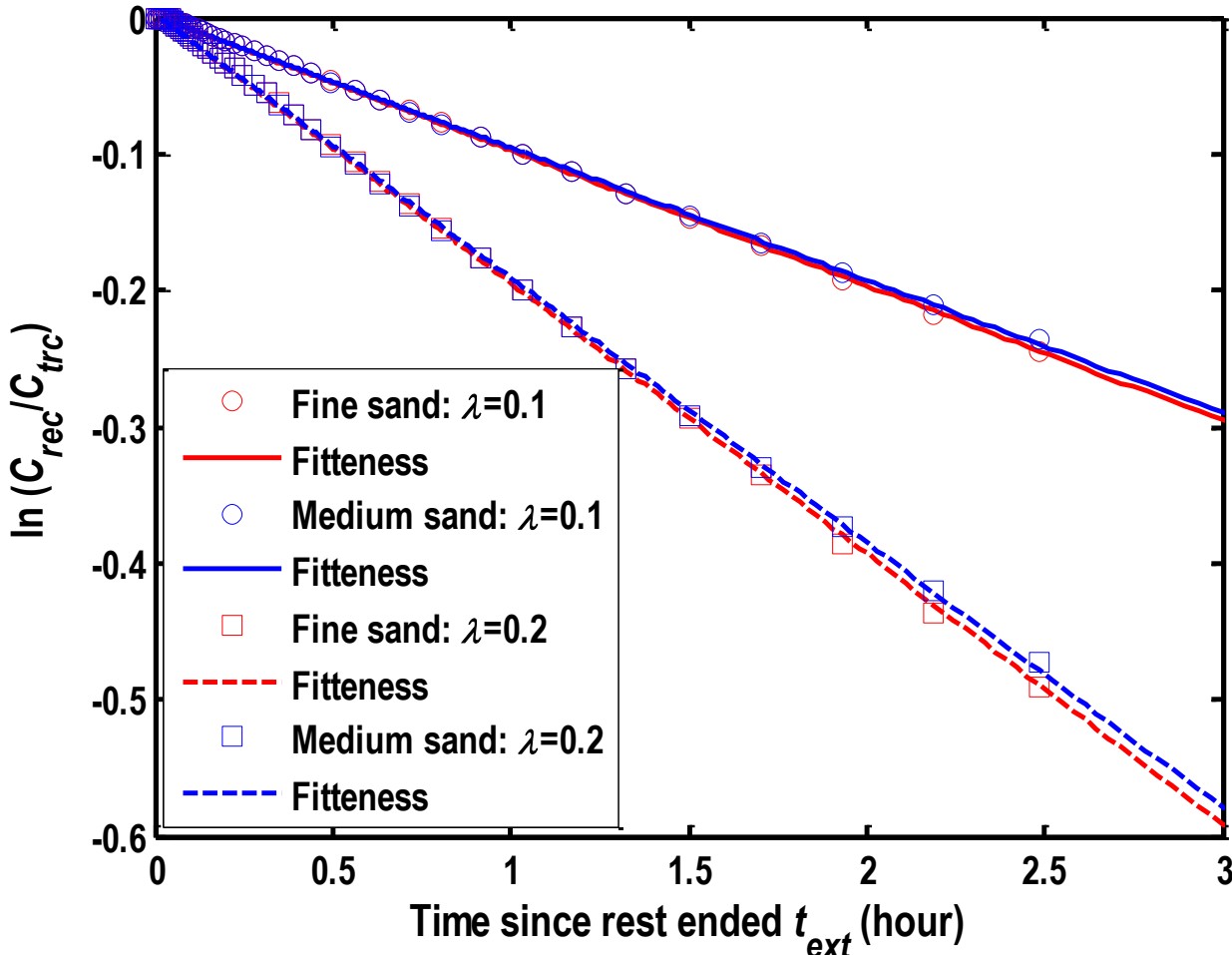

**Figure 9.** Fitness of $\ln\left[C_{rec}/C_{tra}\right] \sim t^{*}$ produced by the numerical solution of this study with the first order reaction in the different porous media.

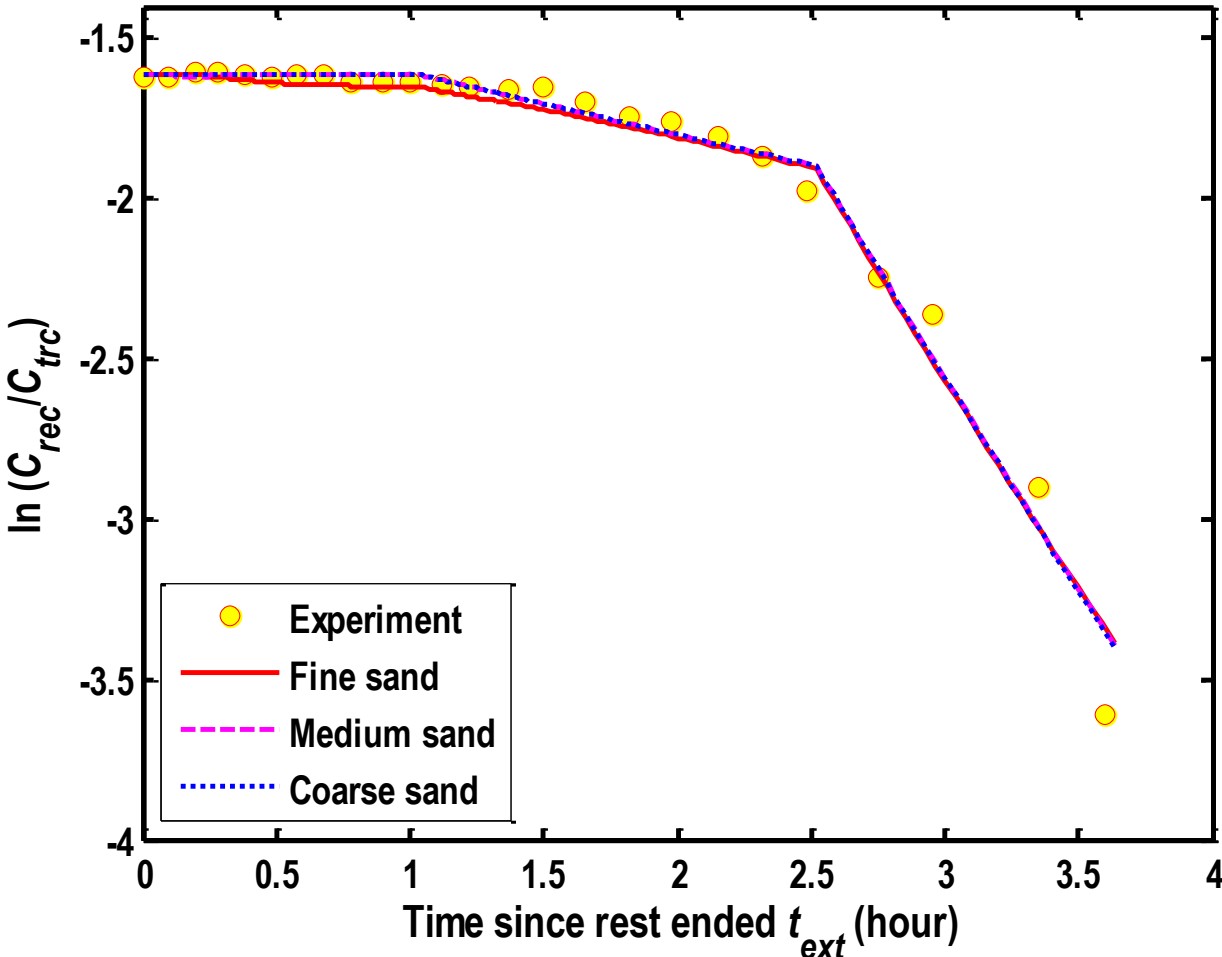

**Figure 10.** Computed $\ln[C_{rec}/C_{tra}] \sim t^*$ by the model of this study using a piecewise linear function to describe the nonlinear chemical reactions.

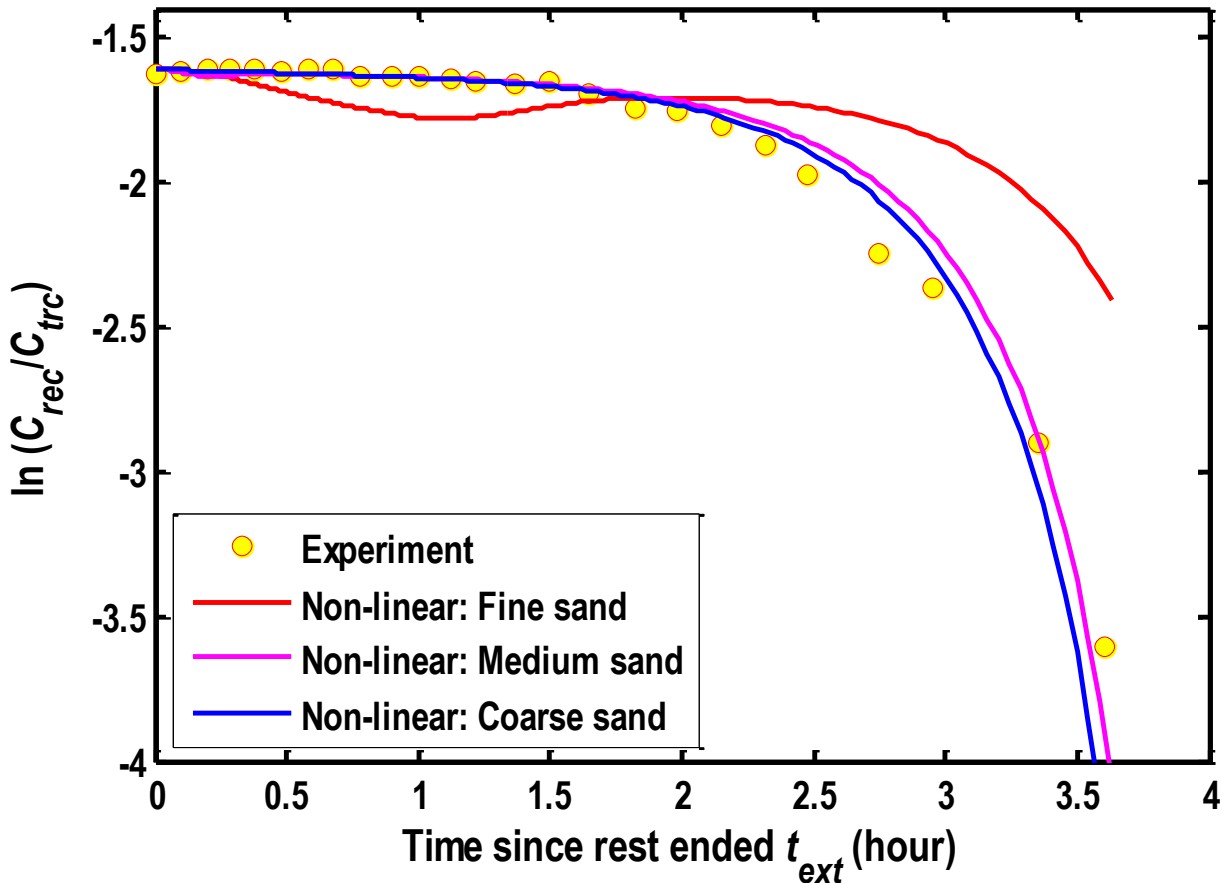

**Figure 11.** Computed $\ln[C_{rec}/C_{tra}] \sim t^{*}$ by the model of this study using a nonlinear function to describe the nonlinear chemical reactions.

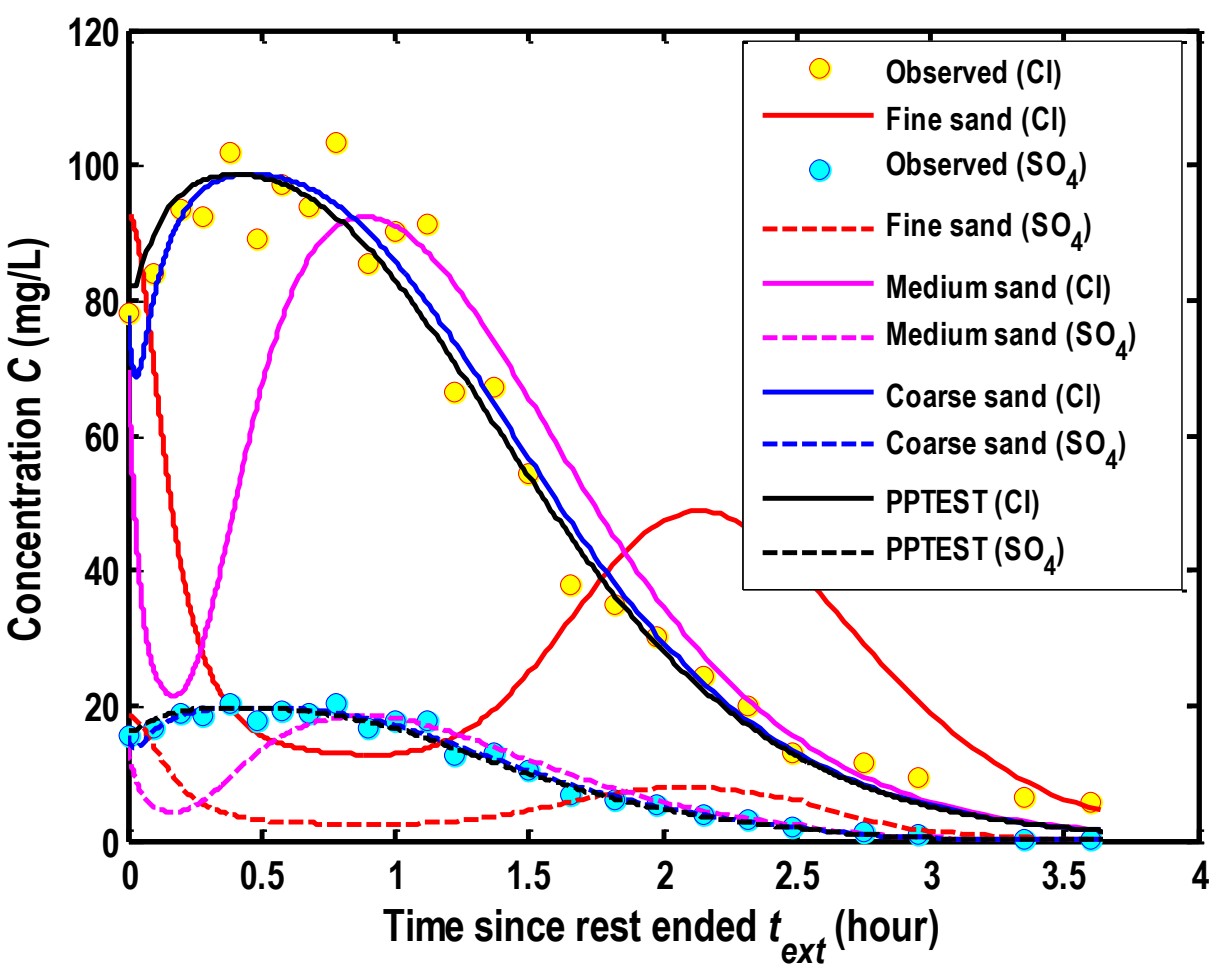

**A**

**Figure 12.** BTCs for the different porous media with a piecewise linear function to describe chemical reactions: A. Cl$^{1-}$ and SO$_4{}^{2-}$ in the aquifer at $r= r_w +0.15$ m, B. Cl$^{1-}$ in the wellbore.

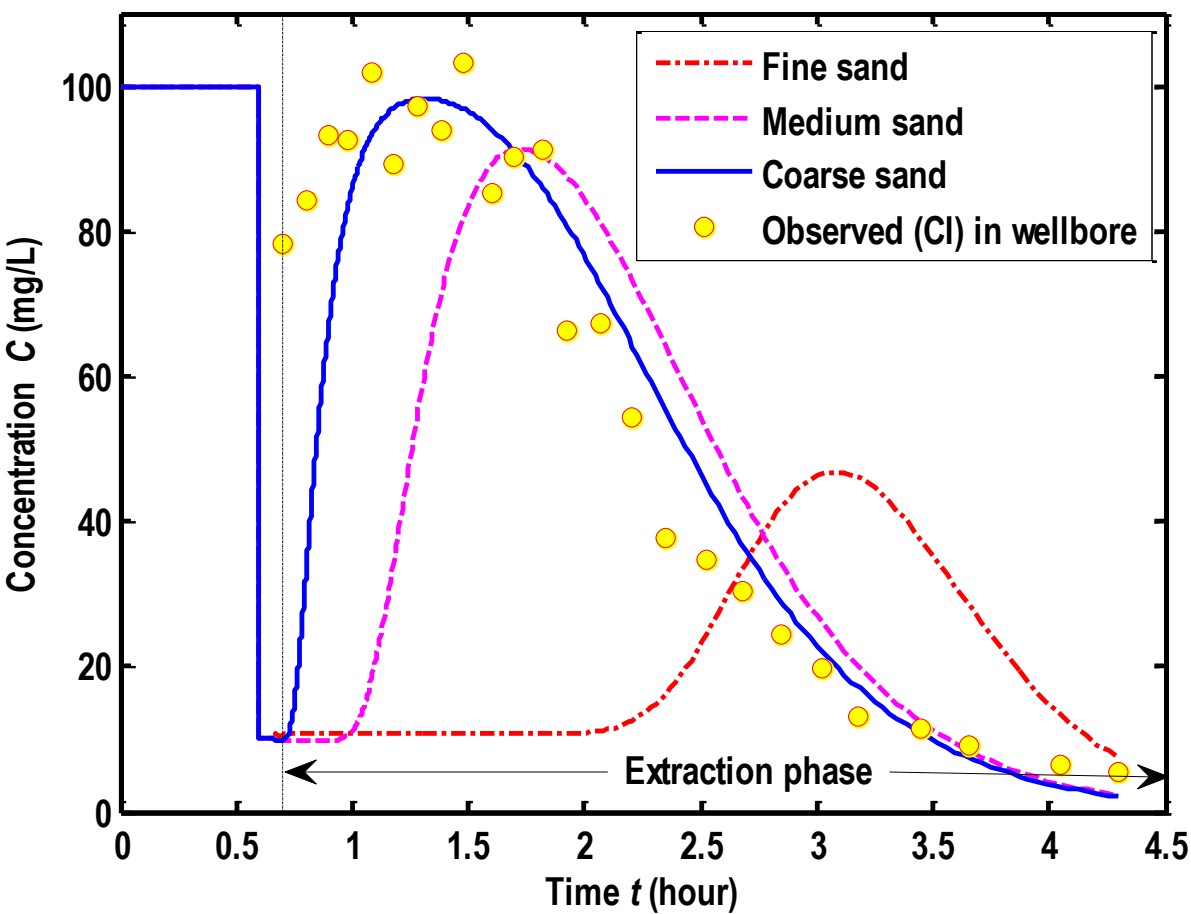

**B**

**Figure 12.** BTCs for the different porous media with a piecewise linear function to describe chemical reactions: A. $Cl^{1-}$ and $SO_4^{2-}$ in the aquifer at $r = r_w + 0.15$ m, B. $Cl^{1-}$ in the wellbore.

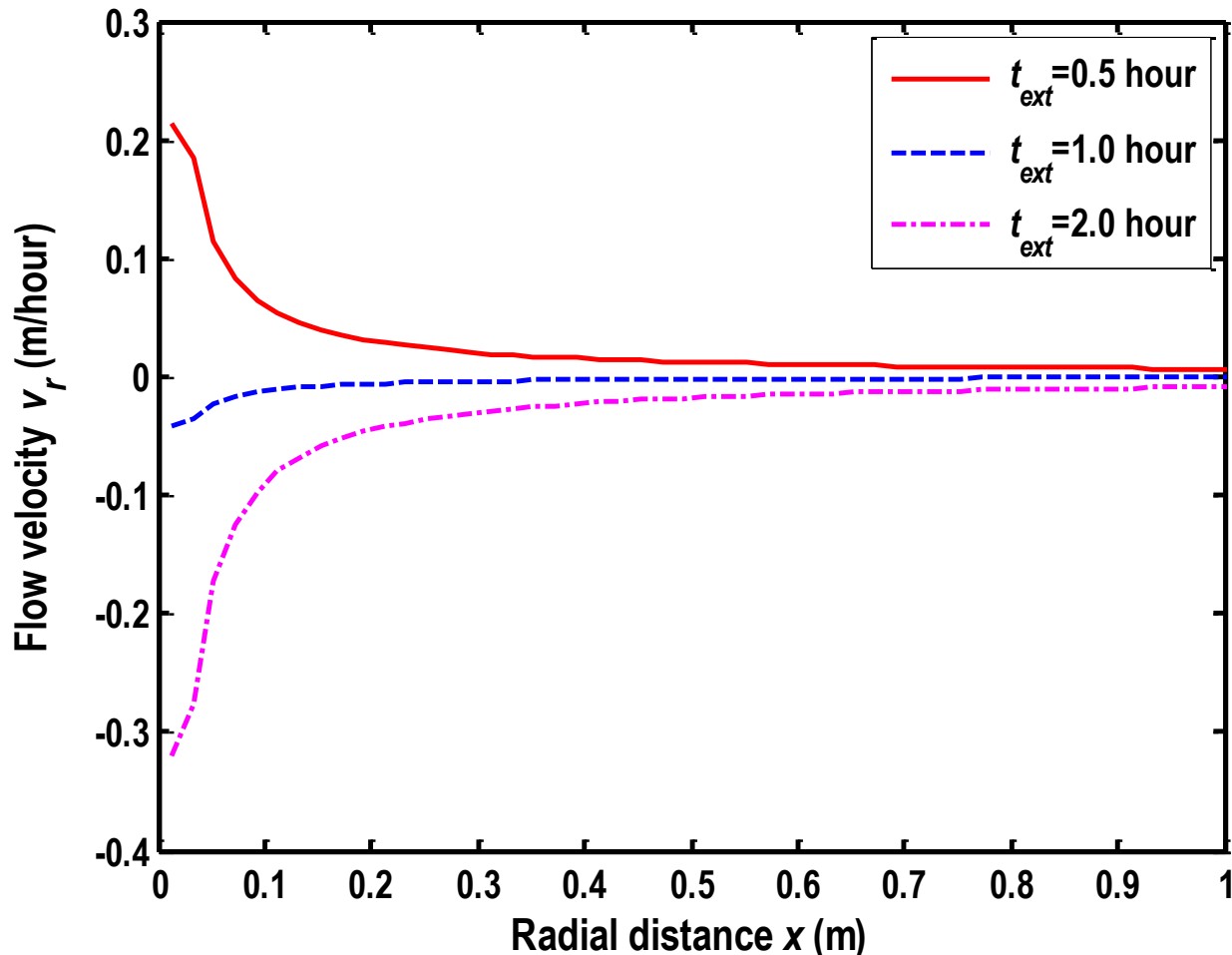

**Figure 13. Spatial distribution of the flow velocity in the extraction phase.**

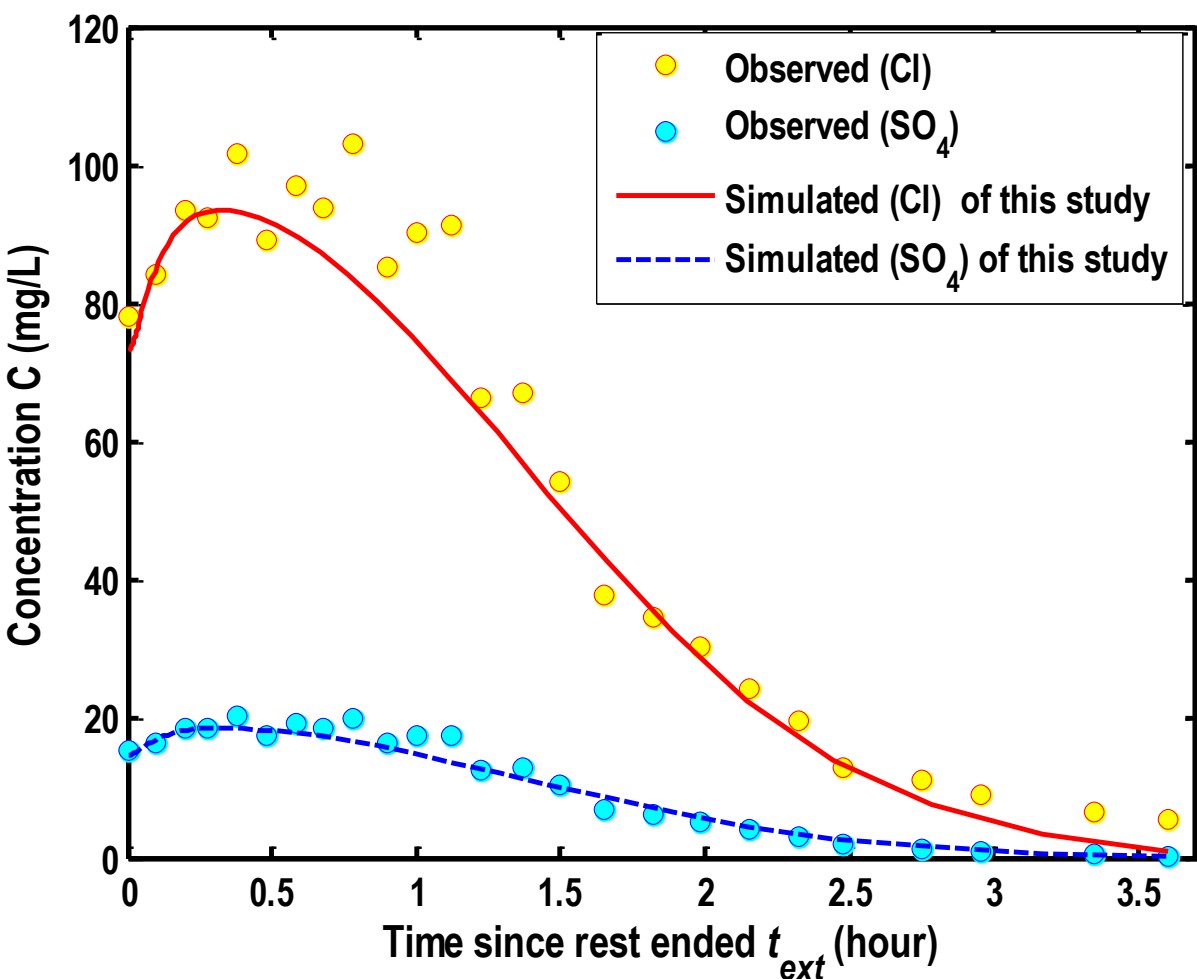

**Figure 14. Fitness of the field SWPP test data by the new model of this study.**

**Table 1. Reaction parameters estimated by linear functions.**

| $K$ (m/day) | $S_s$ (m$^{-1}$) | $\lambda$ (hour) | $\tilde{\lambda}$ (hour) | Intercept of linear function | $\ln\left(\dfrac{1-\exp\left(-\lambda t_{inj}\right)}{\lambda t_{inj}}\right)$ | $\ln\left[\dfrac{C_{rec}^{0}}{C_{trc}^{0}}\right]$ |
|---|---|---|---|---|---|---|
| 0.1 | 0.0001 | 0.1 | 0.0991 | 0.0017 | -0.0299 | 0 |
| 1 | 0.0001 | 0.1 | 0.0970 | 0.0016 | -0.0299 | 0 |
| 0.1 | 0.0001 | 0.2 | 0.1981 | 0.0034 | -0.0594 | 0 |
| 1 | 0.0001 | 0.2 | 0.1939 | 0.0031 | -0.0594 | 0 |

**Nomenclature**

| | |
|---|---|
| $B$ | Aquifer thickness [L] |
| $C_i$ | Aqueous phase concentration of the $i^{th}$ reactive solute [ML$^{-3}$] |
| $C$ | Resident concentration of the aqueous phase to represent $C_i$ in Eq. (1) [ML$^{-3}$] |
| $C_0^{inj}$, $C_0^{cha}$ | Solute concentrations injected into the wellbore during the injection and chase phases [ML$^{-3}$], respectively |
| $C_w^t$, $C_w^{t+\Delta t}$ | Solute concentrations in the wellbore at the time $t$ and $t + \Delta t$ [ML$^{-3}$], respectively |
| $C_{rec}(t^*)$, $C_{tra}(t^*)$ | Reactant concentration and the concentration of a conservative tracer [ML$^{-3}$], respectively |
| $D_r$ | Dispersion coefficient [L$^2$T$^{-1}$] |
| $et^*$ | Time since the end of injection [T] |
| $F_j$ | Monod/Michaelis-Menten kinetics function [dimensionless] |
| $K_r$ | Radial hydraulic conductivity [LT$^{-1}$] |
| $M$ | Number of nodes in discretization of the temporal domain [dimensionless] |
| $\Delta m$ | Mass entering into the well during time interval $\Delta t$ [M] |
| $N$ | Number of the segment for chemical reactions [dimensionless] |
| $N_r$ | Number of nodes in discretization of the spatial domain [dimensionless] |
| $Q$ | Flow rate of the well [L$^3$T$^{-1}$] |
| $r$ | Radius distance from the center of the well [L] |
| $r_i$ | Radial distance of node[L] |
| $r_w$ | Well radius [L] |
| $r_e$ | Distance of outer boundary of the aquifer [L] |
| $s$ | Drawdown [L] |
| $s_w$ | Drawdown inside the wellbore [L] |
| $S_i$ | Solid phase concentration of the $i^{th}$ reactive solute [ML$^{-3}$] |
| $S_s$ | Specific storage of aquifer [L$^{-1}$] |
| $t$ | Time in the SWPP test [T] |

| $t_i$ | Time of node $i$ [T] |
|---|---|
| $t_{inj}, t_{cha}, t_{res}, t_{ext}$ | Durations [T] of the injection, chase, rest, and extraction phases, respectively |
| $t^*$ | Time since the end of injection [T] |
| $t_j^*$ | Times at two ends of segment $j$ [T] |
| $u_r$ | Radial Darcian velocities [LT$^{-1}$] |
| $v_r = u_r/\theta$ | Average radial pore velocity [LT$^{-1}$] |
| $V^t$ | Volume of water in the wellbore at the time $t$ [L$^3$] |
| $\rho_b$ | Bulk density of the aquifer material [ML$^{-3}$] |
| $\theta$ | Porosity [dimensionless] |
| $\mathcal{H}$ | Heaviside step function [dimensionless] |
| $n_j, \lambda_j$ | Orders [dimensionless] and constant [dimensionless] in the temporal segment $j$, respectively |
| $\alpha_r$ | Radial dispersivity [L] |
| $\lambda$ | First-order reaction rate constant [dimensionless] |
| ADE | Advection dispersion equation |
| BTC | Breakthrough curve |
| PPTEST | Solution of Phanikumar and McGuire (2010) |
| SWPP | Single well push pull |