# Peer review of "Reactive Transport with Wellbore Storages in a Single-Well Push-Pull Test"

_Hydrology and Earth System Sciences, 2018_

## Referee Comment (RC1) · Anonymous Referee #1 · 4 Aug 2018

General comments

Based on Wang et al. (2017) and Phanikumar and McGuire (2010), this study proposes an extended model to interpret single-well push-pull (SWPP) tests in homogeneous reservoirs considering transient flow, well-bore storage and reactive transport. The improvements from Wang et al. (2017) concern the change in boundary conditions to better consider the effect of well-bore storage and the extension to reactive transport involving complex chemical reactions based on Phanikumar and McGuire (2010). To show the interest of this extended model, the authors revisited the interpretation of a published SWPP test data set by McGuire et al. (2002) and compared with the model of Phanikumar and McGuire (2010) (available on-line) used to interpret this data set. The authors highlighted that by neglecting the well-bore storage effects and the finite

aquifer hydraulic diffusivity, it underestimates the dispersivity in this case, which is similar to Wang et al. (2017) conclusion. The influence of hydraulic diffusivity on reactive transport appears negligible except for low hydraulic diffusivity and nonlinear chemical reactions. In my opinion, the uniqueness of estimated parameters appears critical and should be addressed. In order to avoid some confusion, a sensitivity analysis should be also performed. Note that the code appears not available from the authors. Please find more details in the following.

1) The travel distance of the tracer from the tested well cannot be estimated from SWPP test without other information. Consequently, the dispersivity and the porosity cannot be estimated separately. The porosity must be fixed using other information in order to estimate the dispersivity from SWPP data set. It seems here that the porosity was assumed but without saying through which means. In addition, by considering the transient nature of the test, which results in two more parameters, it is not clear how dispersivity and the hydraulic properties can be derived independently using the breakthrough curve (BTC) of SWPP test? This raises the question of the uniqueness of the solution and a sensitivity analysis would be helpful.

2) The discussion concerning the impact of the hydraulic diffusivity must be developed by providing an explanation of the mechanisms involved. Even though such discussion might require additional analysis, a simple discussion that only focus on the impact in the BTC appears relatively limited.

3) The authors stated that by neglecting the effects of well-bore storage in the interpretation of the SWPP test, this could result in great errors. As mentioned by the authors, the BTC appears impacted at the early time (i.e., before the peak) and in terms of amplitude. In addition, as demonstrated by the authors, the first-order reaction rate can be interpreted through a simplified solution. Consequently, the effects of well-bore storage seem to impact only the early time of the BTC but not the late-time behavior, which is actually comparable to hydraulic tests. To evaluate properly the impact on the BTC with the assumptions considered in this work, a sensitivity analysis could be actually

very useful. Indeed, even though the late-time behavior was not the subject of this work, mainly because the reservoir was assumed homogeneous, numerous studies have focused on the late-time behavior of BTC in heterogeneous media.

4) As mentioned by the authors, the dispersivities obtained from the previous study and this study are 0.001 m and 0.015 m, respectively. In the user manual provided by Phanikumar, M.S., 2010, another acceptable simulation was performed with a dispersivity of 0.01 m (page 20), which is closed to the dispersivity obtained in this study. By looking at the values of the hydraulic conductivity and the specific storage in this manuscript, they are both perfectly rounded to 1.0 m/h and 1.0 E-5 1/m. Have these values been fixed? Because the authors stated that the hydraulic diffusivity influences the BTC in SWPP test, could the same match be obtained by fixing the dispersivity at 0.001 m and varying only the hydraulic diffusivity? The question of the uniqueness of the solution appears actually very critical. It is also not clear why the BTC was computed at $r = r_w + 0.15$ m and the impact on the result? The comparison with Phanikumar and McGuire (2010) must be completed by providing their interpretation.

Specific comments

Page 3 – line 2: "Similarly, in the rest phase, the concentration is also not 0": Even though some dead zones in some configurations can impact the concentration at the well-bore, note that this effect can be strongly reduced by pumping the water in the well-bore and injecting clean water at the same rate at the end of the injection.

Page 5 – line 20: Why not write the equation (7) and the others using the pore velocity?

Page 5 – line 29: "The assumption works when the chase concentration is zero" This should be normally the case, isn't it?

Page 8 – line 19: "... appears to be accurate and reliable" I would suggest estimating the difference between the two solutions, through for instance an indicator like the mean square residual. This require more points for the solution of Wang et al., 2017.

Page 18 – Figure 2: How do you explain the small oscillation in your solution?

Page 28 – Figure 10: Please add the interpretation of Phanikumar and McGuire (2010) for comparison.

Technical corrections

Page 3 – line 17: remove ", so did Wang et al. (2017)".

Page 3 – line 24: remove "Such an effect is called storage effect of solute transport in this study " as already stated line 2 at the same page.

Page 3 – line 32: remove the extra "the".

Page 9 – line 5 to 10: Please use SI units like m/s for the hydraulic conductivity m2/s for the hydraulic diffusivity or g/l for the concentration.

Page 12 – line 25: Correct "PPTESE" by "PPTEST".

Page 13 – line 22: Remove "In this study" or "of this study".

Page 18 – Figure 3a: Units need to be specified for the concentration. Same comments for the figures 3b, 4 and 10.

Page 23, 24 – Figures 6 and 7: Different units for the time are used while the axes are similar.

Page 27 – Figure 9: Flow velocity is written using qr while it should be vr.

―――――――――――――――

---

## Referee Comment (RC2) · Anonymous Referee #2 · 29 Aug 2018

General comments The authors presented a study to extend the Phanikumar and McGuire (2010) model for the single-well push-pull (SWPP) tests in homogeneous, isotropic aquifers by replacing the oversimplified boundary conditions with more complex conditions to account for wellbore storage effect (the Model). The Phanikumar and McGuire (2010) model is a polar coordinate mathematical model used to interpret SWPP tests involving multi-species reactive transport problems with non-linear reactions. The aim of the study was to reduce the potential errors that may be introduced by ignoring the storage effect in the previous models. The authors verified the accuracy of the Model by comparing the breakthrough covers (BTCs) modeled by the proposed mode with those generated by the Wang et al. (2017) model. The Wang et a. (2017) model is a similar expansion the Phanikumar and McGuire (2010) model
accounting for wellbore storage with respective to groundwater flow. The difference is that the Model accounts for wellbore effects with respect to solute concentration, in addition to flow. Lastly, the authors used the Model to interpret the breakthrough curves (BTCs) of a SWPP field test reported by McGuire et al. (2002) and compared with interpretation of the same BTCs by the Phanikumar and McGuire (2010) model. I have no issues with how the wellbore boundary conditions are formulated. However, I have concerns on the approach the authors taken to verify the Model, which may cast doubts on the practical usefulness of the Model, in specific: 1) Figure 2 shows the comparison between the BTCs modeled by the Model versus those momlded by Wang et al. (2017). But models were used to model the BTCs of hypothetical SWPP tests in three different porous media of fine, medium and coarse sands with typical hydraulic parameters found in textbooks. Figures 3 and 4 illustrate the differences in the BTCs attributable to the wellbore concentration effects. The authors argued the numerical solution of Wang et al. (2017) was chosen to verify the Model because benchmark analytical solutions of the SWPP test with a finite hydraulic diffusivity are not available up to date. An alternative approach is to verify the Model using a solution given by a widely used modeling software, such as MODFOW-SURFACT or FEFLOW (The authors are incorrect to state that commercial numerical software packages are incapable of accurately incorporating the wellbore boundary efforts). 2) To demonstrate it applicability, the Model was used to interpret the BTCs reported in McGuire (2002) in comparison with the Phanikumar and McGuire (2010) model (Figures 6, 7, and 10). Because both models were able to replicate the BTCs, the authors included additional scenarios with varied parameters to demonstrate the differences between the two models. It would be much more compelling to use the field test whose BTCs could not be replicated by the Phanikumar and McGuire (2010) but can be reproduced by the Model. This would alleviate the concern about the necessity of introducing additional complexity into a groundwater model which is known to be subject to parameter uncertainties. In case that such a field test is not available, the authors may consider using a data set modeled using a modeling sofwware such as MODFOW-SURFACT

or FEFLOW. 3) Consider adding a list of acronyms defining the physical meanings of the different symbols. Specific comments 1) Abstract. The abstract should be revised to eliminate the discussion of the details such as the Freundlich, Langmuir and linear sorption models, one-site kinetic sorption model, two-site sorption model, and Monod or Michaelis-Menten kinetics. These are not the core subject of this study. 2) Page 2, line 7. Change to: "the model which is expected to properly represent the physical..." 3) Page 2, line 26-27, change to: "...however, such model only considered wellbore storage effects with respect to groundwater flow, but not solute concentrations." 4) Page 2, line 33, change to: "...concentration of the solute in the wellbore is smaller than that of the original solution..." 5) Page 3, line 11. It is incorrect to state that none of the four software packages could deal with multi-species reactive transport problems with non-linear reactions. Both MODFLOW-SURFACT and FEFLOW can. 6) Page 3, line 29. Define hydraulic diffusivity at its first appearance. Hydraulic diffusivity is a term used mostly in soil physics, not groundwater hydrology. 7) Page 7, line 9. Change to: " ...reactive processes considering wellbore effects not only for groundwater flow but also for solute contrations." 8) Page 9, line 1. It is not clear what does it mean by "Subject to the discharge or recharge of the well,.." Please revise.

Please also note the supplement to this comment:
https://www.hydrol-earth-syst-sci-discuss.net/hess-2018-181/hess-2018-181-RC2-supplement.pdf

––––––––––––––––––––––––––

---

## Author Comment (AC1) · 17 Oct 2018

Response to Referee #1: General Comments: 1. Based on Wang et al. (2017) and Phanikumar and McGuire (2010), this study proposes an extended model to interpret single-well push-pull (SWPP) tests in homogeneous reservoirs considering transient flow, well-bore storage and reactive transport. The improvements from Wang et al. (2017) concern the change in boundary conditions to better consider the effect of well-bore storage and the extension to reactive transport involving complex chemical reactions based on Phanikumar and McGuire (2010). To show the interest of this extended model, the authors revisited the interpretation of a published SWPP test data set by McGuire et al. (2002) and compared with the model of Phanikumar and McGuire (2010) (available on-line) used to interpret this data set. The authors highlighted that

by neglecting the well-bore storage effects and the finite aquifer hydraulic diffusivity, it underestimates the dispersivity in this case, which is similar to Wang et al. (2017) conclusion. The influence of hydraulic diffusivity on reactive transport appears negligible except for low hydraulic diffusivity and nonlinear chemical reactions. In my opinion, the uniqueness of estimated parameters appears critical and should be addressed. In order to avoid some confusion, a sensitivity analysis should be also performed. Note that the code appears not available from the authors. Please find more details in the following. Reply: Implemented. The sensitivity analysis of the flow field on BTCs in wellbore has been conducted (See Section 5.1), and the uniqueness of the BTCs has been investigated (See Section 5.2). The code of this study is free of charge upon request from the authors (See P8 Line 13).

2. The travel distance of the tracer from the tested well cannot be estimated from SWPP test without other information. Consequently, the dispersivity and the porosity cannot be estimated separately. The porosity must be fixed using other information in order to estimate the dispersivity from SWPP data set. It seems here that the porosity was assumed but without saying through which means. In addition, by considering the transient nature of the test, which results in two more parameters, it is not clear how dispersivity and the hydraulic properties can be derived independently using the breakthrough curve (BTC) of SWPP test? This raises the question of the uniqueness of the solution and a sensitivity analysis would be helpful. Reply: Implemented. We have discussed the method to determine the porosity (See P11 Lines 1 - 3). We have discussed how to determine the parameters of hydraulic properties (See P11 Lines 3 - 5). The sensitivity analysis of the flow field on BTCs in wellbore was conducted (See Section 5.1), and the uniqueness of the BTCs was investigated (See Section 5.2).

3. The discussion concerning the impact of the hydraulic diffusivity must be developed by providing an explanation of the mechanisms involved. Even though such discussion might require additional analysis, a simple discussion that only focus on the impact in the BTC appears relatively limited. Reply: Implemented. We have analyzed the impact

of the hydraulic diffusivity on solute transport in the SWPP test (See Section 5).

4. The authors stated that by neglecting the effects of well-bore storage in the interpretation of the SWPP test, this could result in great errors. As mentioned by the authors, the BTC appears impacted at the early time (i.e., before the peak) and in terms of amplitude. In addition, as demonstrated by the authors, the first-order reaction rate can be interpreted through a simplified solution. Consequently, the effects of well-bore storage seem to impact only the early time of the BTC but not the late-time behavior, which is actually comparable to hydraulic tests. To evaluate properly the impact on the BTC with the assumptions considered in this work, a sensitivity analysis could be actually very useful. Indeed, even though the late-time behavior was not the subject of this work, mainly because the reservoir was assumed homogeneous, numerous studies have focused on the late-time behavior of BTC in heterogeneous media. Reply: Implemented. The sensitivity analysis of the flow field on BTCs in wellbore was conducted (See Section 5.1).

5. As mentioned by the authors, the dispersivities obtained from the previous study and this study are 0.001 m and 0.015 m, respectively. In the user manual provided by Phanikumar, M.S., 2010, another acceptable simulation was performed with a dispersivity of 0.01 m (page 20), which is closed to the dispersivity obtained in this study. By looking at the values of the hydraulic conductivity and the specific storage in this manuscript, they are both perfectly rounded to 1.0 m/h and 1.0 E-5 1/m. Have these values been fixed? Because the authors stated that the hydraulic diffusivity influences the BTC in SWPP test, could the same match be obtained by fixing the dispersivity at 0.001 m and varying only the hydraulic diffusivity? The question of the uniqueness of the solution appears actually very critical. It is also not clear why the BTC was computed at r = rw + 0.15 m and the impact on the result? The comparison with Phanikumar and McGuire (2010) must be completed by providing their interpretation. Reply: Implemented. See the following discussion.

Response to the estimation of dispersivity: Although the dispersivities estimated by the

**HESSD**

previous study and this study are almost the same, the method used for the estimation of dispersivity in the previous study is questionable. This is because Phanikumar and McGuire (2010) employed the BTC in the aquifer at r= rw+0.15 m to best fit the observed BTCs in the wellbore (r= rw), as shown in Figure 11A of this study, or Figure 5 of Phanikumar and McGuire (2010), while the authors did not provide a convincing argument why to choose the BTCs in the aquifer to represent the BTCs in the wellbore. Furthermore, the use of "0.15 m" in their analysis appears to be an artifact.

Response to the estimation of hydraulic parameters: The influence of the hydraulic parameters could be ignored. This is because the estimated hydraulic diffusivity ($K_r$ / $S_s$ = 1.0×105 m2/hour) is much greater than the hydraulic diffusivity of the medium sand of 4.17×102 m2/hour, and the BTC is insensitive to the flow field, as shown in Figure 6 of this revised manuscript.

Response to the uniqueness of estimated parameters: When both dispersivity and porosity are unknown, the solutions are non-unique, as shown in Figure 7. However, when the porosity is given in advance, the estimation of dispersivity is unique. In this study, we mainly estimated the value of dispersivity, where the porosity is fixed and comes from the reference of Phanikumar and McGuire (2010). Therefore, the dispersivity is uniquely determined (See P15 Lines 24-26).

Response to the interpretation by Phanikumar and McGuire (2010): Phanikumar and McGuire (2010) did not explain why they choose the BTC at r = rw + 0.15 m to conduct their analysis.

Specific Comments: 1. Page 3 – line 2: "Similarly, in the rest phase, the concentration is also not 0": Even though some dead zones in some configurations can impact the concentration at the well-bore, note that this effect can be strongly reduced by pumping the water in the well-bore and injecting clean water at the same rate at the end of the injection. Reply: Implemented. When the chaser phase is absent, the concentration in the early stage of the rest phase might not be zero (See P3 Lines 1-2).

2. Page 5 – line 20: Why not write the equation (7) and the others using the pore velocity? Reply: Implemented (See Eq.(7)).

3. Page 5 – line 29: "The assumption works when the chase concentration is zero" This should be normally the case, isn't it? Reply: Implemented. In general, the chase concentration is usually set as zero. However, under some circumstances, investigators may use a non-zero concentration for the chase phase. For example, Phanikumar and McGuire (2010) used 10 mg/L for Cl- and 2 g/L for SO42- in their study (See P5 Lines 29-30).

4. Page 8 – line 19: ". . . appears to be accurate and reliable" I would suggest estimating the difference between the two solutions, through for instance an indicator like the mean square residual. This require more points for the solution of Wang et al., 2017. Reply: Implemented (See P8 Line 21).

5. Page 18 – Figure 2: How do you explain the small oscillation in your solution? Reply: Implemented. It shows a small oscillation in the numerical solutions, which might be caused by the numerical errors (See P8 Line 19).

6. Page 28 – Figure 10: Please add the interpretation of Phanikumar and McGuire (2010) for comparison. Technical corrections Reply: Implemented. The observed and computed BTCs by Phanikumar and McGuire (2010) are added in Figures 11A and 11B for comparison (See Figures 11A and 11B).

7. Page 3 – line 17: remove ", so did Wang et al. (2017)". Reply: Removed.

8. Page 3 – line 24: remove "Such an effect is called storage effect of solute transport in this study " as already stated line 2 at the same page. Reply: Removed.

9. Page 3 – line 32: remove the extra "the". Reply: Removed.

10. Page 9 – line 5 to 10: Please use SI units like m/s for the hydraulic conductivity m2/s for the hydraulic diffusivity or g/l for the concentration. Reply: Implemented. The unit of time is revised as hour for the purpose of comparison. This is consistent with

the time unit used in most cited references.

11. Page 12 – line 25: Correct "PPTESE" by "PPTEST". Reply: Implemented (See P14 Line 23).

12. Page 13 – line 22: Remove "In this study" or "of this study". Reply: Removed.

13. Page 18 – Figure 3a: Units need to be specified for the concentration. Same comments for the figures 3b, 4 and 10. Reply: Implemented (See Figures 4a, 4b, 5 and 7).

14. Page 23, 24 – Figures 6 and 7: Different units for the time are used while the axes are similar. Reply: Revised (See Figure 9).

15. Page 27 – Figure 9: Flow velocity is written using qr while it should be vr. Reply: Revised (See Figure 12).

Response to Referee #2: General Comments: 1. The authors presented a study to extend the Phanikumar and McGuire (2010) model for the single-well push-pull (SWPP) tests in homogeneous, isotropic aquifers by replacing the oversimplified boundary conditions with more complex conditions to account for wellbore storage effect (the Model). The Phanikumar and McGuire (2010) model is a polar coordinate mathematical model used to interpret SWPP tests involving multi-species reactive transport problems with non-linear reactions. The aim of the study was to reduce the potential errors that may be introduced by ignoring the storage effect in the previous models. The authors verified the accuracy of the Model by comparing the breakthrough covers (BTCs) modeled by the proposed mode with those generated by the Wang et al. (2017) model. The Wang et al. (2017) model is a similar expansion the Phanikumar and McGuire (2010) model accounting for wellbore storage with respective to groundwater flow. The difference is that the Model accounts for wellbore effects with respect to solute concentration, in addition to flow. Lastly, the authors used the Model to interpret the breakthrough curves (BTCs) of a SWPP field test reported by McGuire et al. (2002) and compared

with interpretation of the same BTCs by the Phanikumar and McGuire (2010) model. I have no issues with how the wellbore boundary conditions are formulated. However, I have concerns on the approach the authors taken to verify the Model, which may cast doubts on the practical usefulness of the Model, Reply: Implemented. We have verified the model of this study using MODFLOW/MT3DMS (See Figure 3).

2. Figure 2 shows the comparison between the BTCs modeled by the Model versus those modeled by Wang et al. (2017). But models were used to model the BTCs of hypothetical SWPP tests in three different porous media of fine, medium and coarse sands with typical hydraulic parameters found in textbooks. Figures 3 and 4 illustrate the differences in the BTCs attributable to the wellbore concentration effects. The authors argued the numerical solution of Wang et al. (2017) was chosen to verify the Model because benchmark analytical solutions of the SWPP test with a finite hydraulic diffusivity are not available up to date. An alternative approach is to verify the Model using a solution given by a widely used modeling software, such as MODFOW-SURFACT or FEFLOW (The authors are incorrect to state that commercial numerical software packages are incapable of accurately incorporating the wellbore boundary efforts). Reply: Implemented.

Response to verify the model of this study: We used MODFLOW/MT3DMS to test the model of this study (See Figure 3).

Response to errors in MODFOW-SURFACT or FEFLOW: It is necessary to test the accuracy of the new models against the commercial numerical software packages, like MODFLOW/MT3DMS. Unfortunately, the current three-dimensional models in the MODFLOW/MT3DMS may create some errors in describe the solute transport in the wellbore-confined aquifer. The errors come from an assumption that the water volume in the wellbore is computed by a product of wellbore cross section and the aquifer thickness, which is incorrect. The actual water volume in the wellbore should be computed by a product of wellbore cross section and the water level in the wellbore (See Supplemental Materials for detailed explanation). The models of solute transport in

MODFLOW/MT3DMS are the same with the models in MODFOW-SURFACT or FE-FLOW.

3. To demonstrate it applicability, the Model was used to interpret the BTCs reported in McGuire (2002) in comparison with the Phanikumar and McGuire (2010) model (Figures 6, 7, and 10). Because both models were able to replicate the BTCs, the authors included additional scenarios with varied parameters to demonstrate the differences between the two models. It would be much more compelling to use the field test whose BTCs could not be replicated by the Phanikumar and McGuire (2010) but can be reproduced by the Model. This would alleviate the concern about the necessity of introducing additional complexity into a groundwater model which is known to be subject to parameter uncertainties. In case that such a field test is not available, the authors may consider using a data set modeled using a modeling software such as MODFOW-SURFACT or FEFLOW. Reply: As mentioned in reply to comment #2 of this reviewer, because of the problem of MODFLOW/MT3DMS for computing the wellbore water volume, it is not advisable to using a synthetic data set generated using a modeling software such as MODFOW-SURFACT or FEFLOW (See Supplementary Materials for detailed explanation of this issue).

4. Consider adding a list of acronyms defining the physical meanings of the different symbols. Reply: Implemented (See Nomenclature).

Specific Comments: 1. Abstract. The abstract should be revised to eliminate the discussion of the details such as the Freundlich, Langmuir and linear sorption models, one-site kinetic sorption model, two-site sorption model, and Monod or Michaelis-Menten kinetics. These are not the core subject of this study. Reply: Implemented (See Abstract).

2. Page 2, line 7. Change to: "the model which is expected to properly represent the physical..." Reply: Implemented (See P2 Line 5).

3. Page 2, line 26-27, change to: "...however, such model only considered wellbore

storage effects with respect to groundwater flow, but not solute concentrations." Reply: Implemented (See P2 Lines 26-27).

4. Page 2, line 33, change to: "...concentration of the solute in the wellbore is smaller than that of the original solution..." Reply: Implemented (See P2 Line 29).

5. Page 3, line 11. It is incorrect to state that none of the four software packages could deal with multispecies reactive transport problems with non-linear reactions. Both MODFLOW-SURFACT and FEFLOW can. Reply: Such a statement has been deleted.

6. Page 3, line 29. Define hydraulic diffusivity at its first appearance. Hydraulic diffusivity is a term used mostly in soil physics, not groundwater hydrology. Reply: Implemented (See P2 Lines 21-24).

7. Page 7, line 9. Change to: " ...reactive processes considering wellbore effects not only for groundwater flow but also for solute contrations." Reply: Implemented (See P7 Line 10).

8. Page 9, line 1. It is not clear what does it mean by "Subject to the discharge or recharge of the well,.." Please revise. Reply: Implemented (See P9 Lines 9-10).

Please also note the supplement to this comment:
https://www.hydrol-earth-syst-sci-discuss.net/hess-2018-181/hess-2018-181-AC1-supplement.pdf

---

## Author Response (AR2)

College of Geosciences

Department of Geology & Geophysics, College Station, TX 77843-3115
Hongbin Zhan, Ph.D., P.G.
Professor of Geology and Geophysics
Professor of Water Management and Hydrological Sciences
Professor of Energy Institute of Texas A&M University
Holder of Endowed Dudley J. Hughes Chair in Geology and Geophysics
Tel: (979) 862-7961, Fax: (979) 845-6162, Email: zhan@geos.tamu.edu
http://geoweb.tamu.edu/zhan

[Figure]

Memorandum

To: Dr. Monica Riva, Editor of Hydrology and Earth System Sciences

Subject: Revision of Paper hess-2018-181
* * *
**Dear Editor:**

Upon your recommendation, we have carefully revised Paper hess-2018-181 entitled "Reactive Transport with Wellbore Storages in a Single-Well Push-Pull Test" after considering all the comments made by the reviewers. The following is the point-point response to all the comments.

**Response to Referee #1:**

1. The Authors have seriously taken in consideration the main comments of the reviews and provided a stronger revised manuscript. Some technical-very minor changes are suggested.

**Reply:** Implemented. We have carefully revised the manuscript.

**Response to Referee #1:**

**General Comments:**

1. I reviewed the first version of this manuscript. Since all important suggestions have been taken into account through additional analysis, I think that this manuscript can be published..

**Reply:** Thanks a lot.

**Specific Comments:**

1. Page 2 - Line 19: replace "fracture" by "fractured".

**Reply:** Implemented (See P2 Line 19).

2. Page 3 - Line 29: replace "could be" by "is".

**Reply:** Implemented (See P3 Line 29).

108 Halbouty
3115 TAMU
College Station, TX 77843-3115

Tel. 979.845-2451 Fax 979.845-61627
Geoweb.tamu.edu

3. Page 14 - Lines 6 to 9: I agree with this conclusion. However, since pressure has to be measured during SWPP tests, the authors could suggest estimating permeability of the tested zone using for instance the pumping phase.

**Reply:** Implemented (See P14 Lines 14 - 19).

**Response to Referee #2:**

**General Comments:**

1. I appreciate the authors' effort to address my comments, including the comparison of the new model results against the results simulated using 3-D numerical model package MODFLOW/MT3DMS.

**Reply:** Thanks a lot.

**Specific Comments:**

1. I think the authors are incorrect with respect to the calculation of the water volume in the wellbore storage in MODFLOW-SURFACT and FEFLOW. As far as I know, MODFLOW-SURFACT includes a fracture-well package (FWL4 and FWL5) to overcome the problems in the original MODFLOW Well package. The FWL4 and FWL5 packages calculate the water volume using simulated heads, not aquifer thicknesses. The authors can refer to MODFLOW-SURFACT manual, Vol I, Section 3.2, Eq. 24 for details. FEFLOW also has a similar package, referred to as Discrete-feature to simulate a pumping/extraction well, if one chooses to do so. Additionally, with a FEFLOW model, the model mesh can be highly discretized to accurately represent well dimensions using a subset of elements (in centers). The modeler can assign a porosity of unit for those elements representing the wells, rather than assuming the same porosity of the surrounding materials.

**Reply:** Thank you very much for bringing this to our attention, and we have corrected the relevant sentences (See P10 Lines 3 - 6 and Lines 10 - 25). In the future, we will conduct a comprehensive comparative investigation of the method proposed in this study and those of MODFLOW-SURFACT and FEFLOW for understanding the effects of well mixing and wellbore storage for both flow and transport processes involving an aquifer-well system.

If you have any further questions about this revision, please contact me.

Sincerely Yours,

Hongbin Zhan, PhD, PG.